# Post-meiotic DNA double-strand breaks occur in *Tetrahymena*, and require Topoisomerase II and Spo11

Takahiko Akematsu[1]*, Yasuhiro Fukuda[2,3,4], Jyoti Garg[5], Jeffrey S Fillingham[6], Ronald E Pearlman[5], Josef Loidl[1]

[1]Department of Chromosome Biology, University of Vienna, Vienna, Austria; [2]Department of Biodiversity Science, Tohoku University, Oosaki, Japan; [3]Division of Biological Resource Science, Tohoku University, Oosaki, Japan; [4]Graduate School of Agricultural Science, Tohoku University, Oosaki, Japan; [5]Department of Biology, York University, Toronto, Canada; [6]Department of Chemistry and Biology, Ryerson University, Toronto, Canada

**Abstract** Based on observations of markers for DNA lesions, such as phosphorylated histone H2AX (γH2AX) and open DNA ends, it has been suggested that post-meiotic DNA double-strand breaks (PM-DSBs) enable chromatin remodeling during animal spermiogenesis. However, the existence of PM-DSBs is unconfirmed, and the mechanism responsible for their formation is unclear. Here, we report the first direct observation of programmed PM-DSBs via the electrophoretic separation of DSB-generated DNA fragments in the ciliate *Tetrahymena thermophila*. These PM-DSBs are accompanied by switching from a heterochromatic to euchromatic chromatin structure in the haploid pronucleus. Both a topoisomerase II paralog with exclusive pronuclear expression and Spo11 are prerequisites for PM-DSB induction. Reduced PM-DSB induction blocks euchromatin formation, characterized by histone H3K56 acetylation, leading to a failure in gametic nuclei production. We propose that PM-DSBs are responsible for histone replacement during the reprogramming of generative to undifferentiated progeny nuclei.

*For correspondence: takahiko.akematsu@univie.ac.at

**Competing interests:** The authors declare that no competing interests exist.

## Introduction

DNA double-strand breaks (DSBs) represent one of the greatest threats to genome integrity. Nevertheless, deliberate DSB induction is necessary for reshuffling genes or DNA sequences. The most common example in eukaryotes is the induction of meiotic DSBs (*Keeney et al., 1997*), and others include the somatic recombination of immunoglobulin genes (*Stavnezer et al., 2008*), mating-type switching in yeast (*Haber, 2012*), and inducing antigenic variation in *Trypanosoma* (*McCulloch et al., 2015*). Post-meiotic DSBs (PM-DSBs) are a novel type of programmed DSBs that are claimed to occur during spermiogenesis in animals, including humans (*Marcon and Boissonneault, 2004*) and *Drosophila* (*Rathke et al., 2007*). Markers of DNA lesions such as phosphorylated histone H2AX (γH2AX) foci and terminal deoxynucleotidyl transferase dUTP nick end-labeling (TUNEL)-positive signals are found in the nuclei of elongating spermatids (*Marcon and Boissonneault, 2004*; *Meyer-Ficca et al., 2005*; *Leduc et al., 2008*). Moreover, both poly ADP-ribose (PAR) formation, a known DNA damage response (*Meyer-Ficca et al., 2005*) and DNA polymerase activity, characteristic of DNA repair synthesis, have been detected in these cells (*Leduc et al., 2008*). PM-DSBs have been implicated in eliminating free DNA supercoils formed during canonical histone withdrawal to ensure protamine deposition onto untangled DNA (*Marcon and Boissonneault, 2004*; *Laberge and Boissonneault, 2005*; *Rathke et al., 2014*). However, the nature of these lesions

(whether DSBs or single-strand nicks) and the mechanism by which they are formed have not been fully elucidated.

As the TUNEL signal is greatly diminished in rabbit spermatids in the presence of the type II topo-isomerase (Top2) inhibitor, etoposide (*Laberge and Boissonneault, 2005*), which forms a ternary complex with DNA and Top2 (*Pommier et al., 2010*), the catalytic activities of Top2-related proteins have been considered responsible for inducing post-meiotic DNA lesions. Vertebrates encode two closely related *TOP2* genes termed $\alpha$ and $\beta$, which are differentially regulated during cell growth and cannot substitute for each other (*Roca, 2009*). Top2$\alpha$ is only expressed in proliferating cells, where it is involved in mitosis-related events such as DNA replication, chromosome condensation and decondensation, and sister chromatid segregation (*Woessner et al., 1991*; *Grue et al., 1998*). In contrast, Top2$\beta$ is mainly expressed in terminally differentiated cells and is therefore thought to have a non-mitotic function (*Austin and Marsh, 1998*; *Linka et al., 2007*). In cultured human cells, *Ju et al. (2006)* uncovered a role for Top2$\beta$ in catalyzing site-specific DSB formation within some gene promoters, leading to local changes in chromatin architecture and transcriptional activation. In mouse spermiogenesis, *Leduc et al. (2008)* demonstrated that Top2$\beta$ was exclusively expressed in elongating spermatids, where it colocalized with $\gamma$H2AX foci. Moreover, these authors showed that tyrosyl-DNA phosphodiesterase 1 (Tdp1), known to dissociate the covalent bonds between Top2 and DNA (*Nitiss et al., 2006*; *Murai et al., 2012*), was expressed in the same cells. These findings strongly suggest that Top2$\beta$ produces transient DSBs in haploid chromosomes to support chromatin remodeling.

In addition to Top2$\beta$, Spo11, the predominant meiotic DSB inducer (*Keeney, 2008*), has recently been implicated in PM-DSB formation. *Gouraud et al. (2013)* reported that the elongating sperma-tids of mice express much higher levels of *SPO11* transcripts compared with *TOP2$\beta$* transcripts. Since Spo11 has a similar structure to prokaryotic Top2 (also called Top6) and produces DSBs in the same manner as Top2 by forming phosphotyrosine linkages to 5' strand termini on both sides of a DSB (*Neale et al., 2005*), Spo11 may also be involved in inducing PM-DSBs.

Using the unicellular ciliated protist *Tetrahymena thermophila* (hereafter referred to as *Tetrahymena*) as a model biological system, we demonstrate that PM-DSBs are formed in non-metazoan organisms and report the first compelling genetic evidence for the mechanism of PM-DSB induction. The availability of gene manipulation methods (*Chalker, 2012*), sequenced genomes (*Eisen et al., 2006*; *Hamilton et al., 2016*), and gene expression data (*Miao et al., 2009*) make *Tetrahymena* an ideal model for studying fundamental cellular and molecular processes. A remarkable and virtually unique feature of ciliates (including *Tetrahymena*) is that they stably maintain spatially and function-ally differentiated germline and somatic nuclear genomes within a single cytoplasm (*Orias et al., 2011*). The transcriptionally inert diploid germline genome, housed within the micronucleus (MIC), stores the genetic information for sexual progeny, while the polyploid somatic genome, housed within the macronucleus (MAC), is involved in active transcription. Both types of nuclei derive from a single zygotic nucleus during sexual reproduction. The MAC anlagen then undergo large-scale genome rearrangement and amplification processes that remove the internal eliminated sequences, representing about 40% of the genome (*Mochizuki, 2010*, *2012*; *Noto et al., 2015*). The MAC genotype governs the phenotypes of both the MAC and MIC (*Orias et al., 2011*). Owing to its nuclear dualism, *Tetrahymena* is an ideal tool to address questions about genes that have important post-meiotic functions but are not manipulable in other model systems because they are essential for germline cell homeostasis.

In this study, we investigate the functions of *Tetrahymena TOP2$\beta$* and *SPO11* orthologs in haploid MICs (pronuclei) after completing meiosis and show their involvement in PM-DSB formation. We also provide evidence that DNA repair is concomitant with the incorporation of newly synthesized histone H3 into pronuclei. Our data suggest that *Tetrahymena* undergoes a spermiogenesis-like post-meiotic stage, in which Top2-related proteins induce transient DSBs followed by a dynamic change in the chromatin structure of gametic nuclei prior to fertilization.

## Results

### Novel γH2AX localization in pronuclei

*Figure 1* illustrates the process of *Tetrahymena* cell mating (known as conjugation; described in *Cole and Sugai, 2012*). Mating is initiated by the interaction of cells of different mating types, followed by meiotic prophase in both mating partners, during which MICs stretch out to form bivalent chromosomes without synaptonemal complex formation (*Loidl et al., 2012*). Finally, two consecutive meiotic divisions (anaphase I and II) take place to form four identical haploid pronuclei: one of these (the selected pronucleus) undergoes an additional mitosis event (gametogenic mitosis) to produce gametic nuclei, whereas the other three (unselected) pronuclei eventually undergo autophagic degradation (*Liu and Yao, 2012*). After they are reciprocally exchanged between mating partners, each gametic nucleus forms a zygotic nucleus by karyogamy, corresponding to fertilization in metazoa. Soon thereafter, the zygotic nucleus undergoes two consecutive mitotic divisions (post-zygotic mitoses) to produce four identical anlagen. Two of these are distributed to the anterior region of the cytoplasm, whereby they differentiate into the progeny MACs; the remaining two posterior anlagen become progeny MICs. Once the progeny MACs begin to develop, the parental MAC (pMAC) becomes transcriptionally inactive and is selectively eliminated from the cytoplasm via autophagy (*Akematsu et al., 2010*, *Akematsu et al., 2012*, *2014*). Mating is terminated once the progeny MACs develop. One of the progeny MICs is resorbed, while the remaining MIC undergoes replication prior to the first cell division. The progeny MACs are then distributed to the daughter cells. Finally, four progeny cells are produced from a mating cell pair (*Figure 1*).

We discovered H2AX phosphorylation in post-meiotic pronuclei by γH2AX immunostaining (*Figure 2A*). The post-meiotic stage is distinct from the other two meiotic stages in which γH2AX foci are known to be formed, namely (1) during DSB formation in the elongating meiotic prophase MIC (*Mochizuki et al., 2008*; *Papazyan et al., 2014*) and (2) during DNA elimination in the

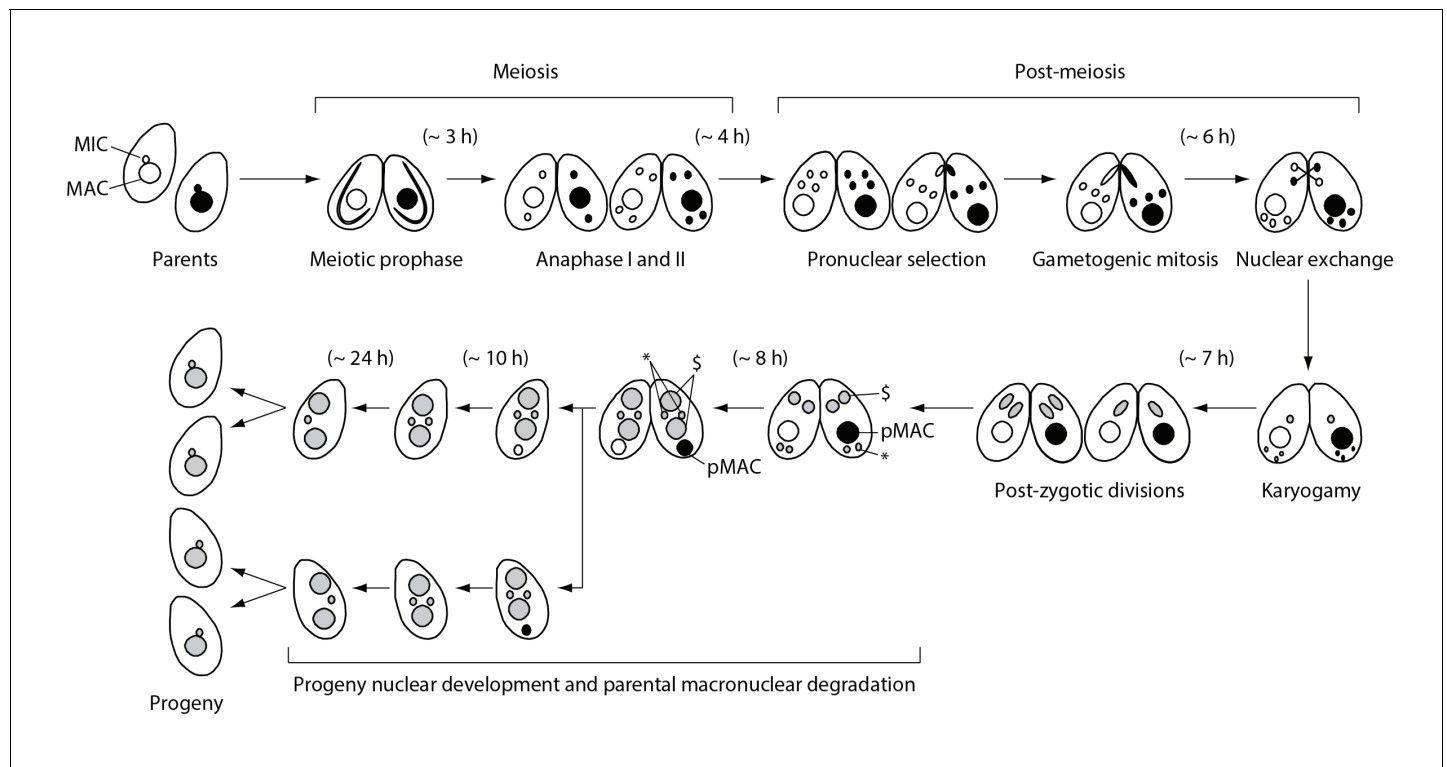

**Figure 1.** Nuclear events during wild-type *Tetrahymena* mating. When starved cells of different mating types are mixed, they start mating and meiosis, and produce sexual progeny. MAC-macronucleus; MIC-micronucleus; $-progeny macronuclear anlagen; *-progeny MICs; pMAC-degrading parental macronucleus. Time (h) after mixing of cells is indicated.

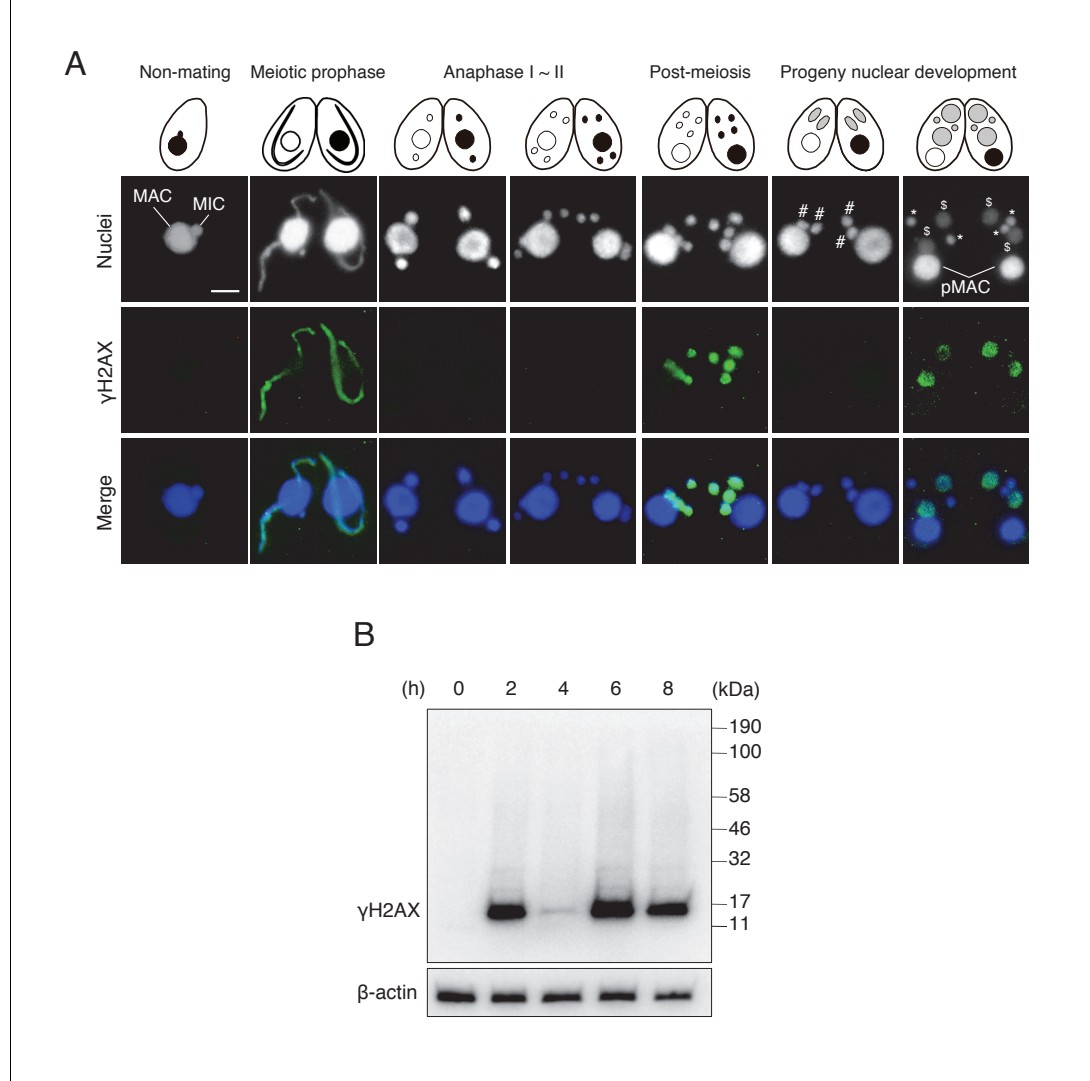

**Figure 2.** γH2AX foci formation, indicating histone H2AX phosphorylation, in *Tetrahymena* mating. (**A**) γH2AX foci appear at three distinct time points—in elongating micronuclei at meiotic prophase—in pronuclei at the post-meiotic stage; and in the developing anlagen—but do not appear in parental MACs throughout mating. Cartoons illustrate the corresponding stages from *Figure 1*. Scale bar denotes 10 µm. MAC-macronucleus; MIC-micronucleus; #-differentiating zygotic nuclei; $-progeny macronuclear anlagen; *-progeny micronuclei; pMAC-degrading parental macronuclei. (**B**) Western blot analysis of γH2AX from different time points during mating. *β*-actin was the loading control. See also *Figure 2—figure supplement 1*.

The following figure supplement is available for figure 2:

**Figure supplement 1.** Subcellular localization of C-terminally GFP-tagged H2AX during the pre-zygotic stages.

developing MAC anlagen (*Figure 2A*) (*Song et al., 2007*). Western blotting of mating cell proteins from different time points showed a consistent result: a single band of about 15 kDa (the size of H2AX) that transiently appeared at 2 hr (meiotic prophase), 6 hr (post-meiotic stage), and 8 hr (MAC development stage) after induction of meiosis (*Figure 2B*). Moreover, a parentally expressed H2AX (encoded by TTHERM_00790790)-GFP fusion protein localized to the MIC at meiotic prophase and to post-meiotic pronuclei (*Figure 2—figure supplement 1*), confirming that pronuclear γH2AX immunostaining was not due to disappearance of H2AX.

In one of the four pronuclei, γH2AX fluorescence disappeared after about 30 min (*Figure 3A/a/ A′*); only this nucleus underwent gametogenic mitosis (*Figure 3B/b/B′*). In contrast, the γH2AX signal persisted in the unselected pronuclei, which relocated to the posterior region of the cell were

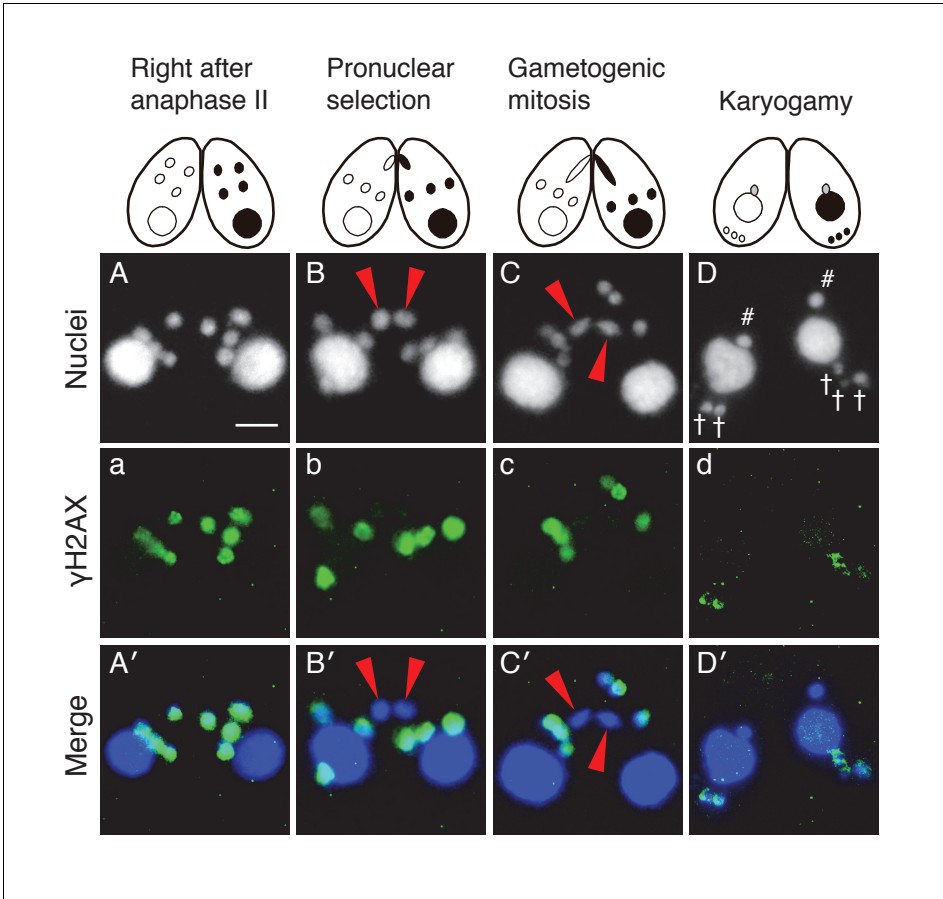

**Figure 3.** Post-meiotic γH2AX and its relation to gametogenic mitosis. (**A/a/A′**) γH2AX is formed in all four pronuclei after completion of meiosis. (**B/b/B′**) A pronucleus (red arrowheads) becomes immunonegative for γH2AX. (**C/c/C′**) Soon thereafter, the pronucleus without γH2AX (red arrowheads) is selected to undergo gametogenic mitosis. (**D/d/D′**) The unselected pronuclei, in which γH2AX (†) persists, are degraded at the posterior region of the cytoplasm and eventually disappear by the karyogamy stage. #-zygotic nuclei. Scale bar denotes 10 μm. See also *Figure 3—figure supplements 1* and *2*.

The following figure supplements are available for figure 3:

**Figure supplement 1.** Possible involvement of NHEJ in PM-DSB repair.

**Figure supplement 2.** Possible involvement of recombination protein Rad51 in PM-DSB repair.

degraded (*Figure 3C/c/C′*). Since H2AX dephosphorylation is an established marker of repaired DNA (*Chowdhury et al., 2005*; *Keogh et al., 2006*), the post-meiotic stage might involve DNA damage formation in all pronuclei, followed by DNA repair only in the selected pronucleus. Indeed, DNA repair markers such as DNA-dependent protein kinase catalytic subunit (DNA-PKcs, a factor involved in DNA repair by non-homologous end joining [NHEJ]) and Rad51 (a protein involved in recombinational repair) are expressed in the selected pronucleus (*Figure 3—figure supplements 1* and *2*).

## Top2 and its relation to post-meiotic γH2AX formation

Ciliates including *Tetrahymena* encode multiple *TOP2* genes (*Figure 4A*). We found that *Tetrahymena* has two closely related Top2 isoforms (encoded by TTHERM_00456750 and TTHERM_00825440; *Figure 4A*), both of which contain domains characteristic of mammalian Top2α

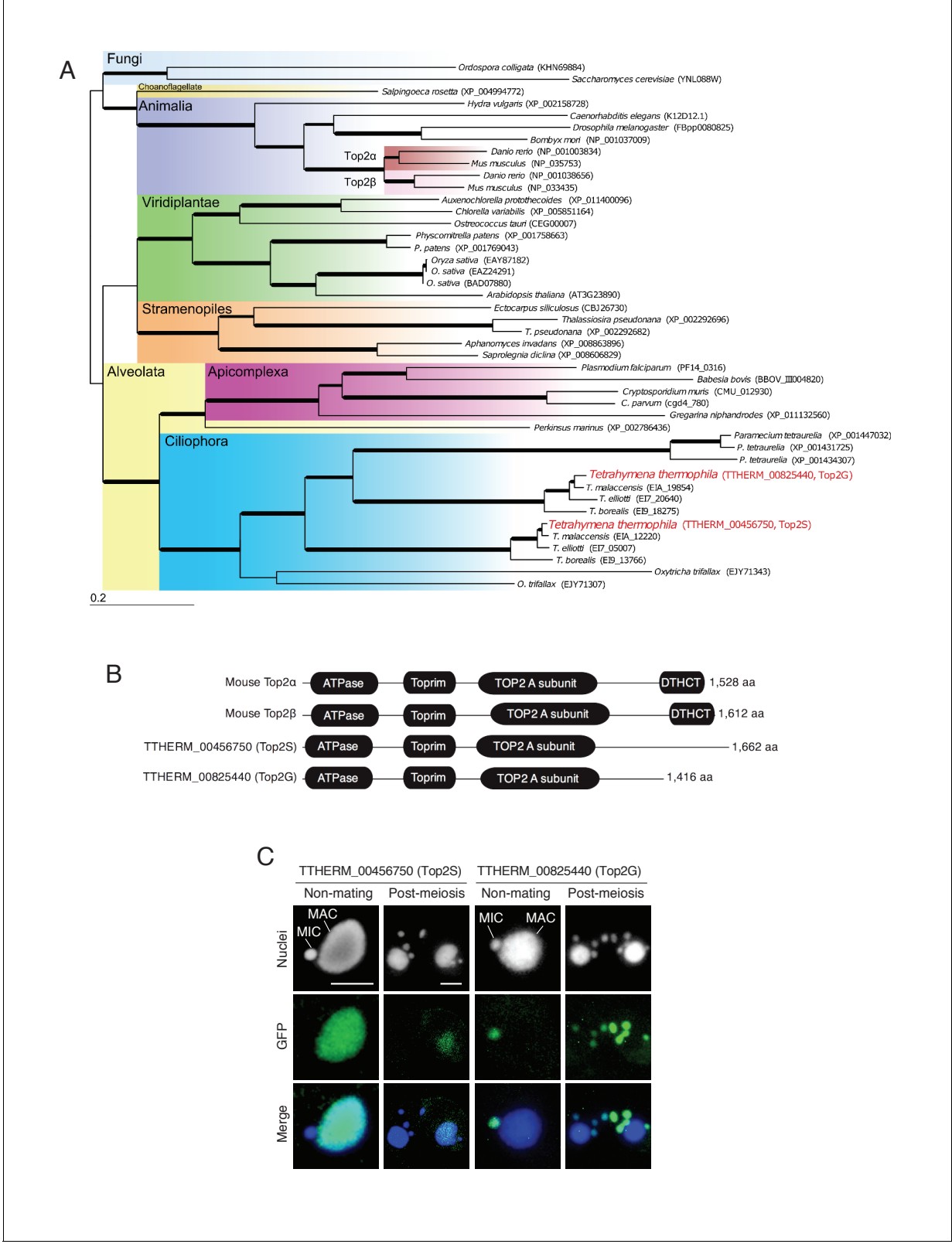

**Figure 4.** Conservation and expression pattern of *Tetrahymena* topoisomerase 2 (Top2) homologs. (**A**) Phylogenetic tree of Top2 homologs. The tree was reconstructed with the maximum likelihood method (see Materials and methods). Accession numbers indicate protein sequences obtained from GenBank. Branch width represents the bootstrap value. Scale bar denotes 0.2 expected amino acid substitutions per site. (**B**) Comparison of conserved domains between mouse and *Tetrahymena* Top2 homologs. (**C**) Subcellular localization of C-terminally GFP-tagged *Tetrahymena* Top2 paralogs. Based

*Figure 4 continued on next page*

*Figure 4 continued*

on their localization, TTHERM_00456750p and TTHERM_00825440p were designated Top2S (Top2 in the somatic nucleus) and Top2G (Top2 in germline nucleus), respectively. In mating pairs, Top2G-GFP expressed in one of the mating partners (right) migrates to the pronucleus of the untagged cell (left), causing a weak signal. MAC-macronucleus, MIC-micronucleus. Scale bars denote 10 μm.

and β: an ATPase domain, a Toprim (topoisomerase–primase) domain, and an N-terminal Top2 A subunit domain (*Figure 4B*). However, both lack the DTHCT domain found in the C-termini of DNA gyrase B, topoisomerase IV (Top4), and H⁺-ATPase proteins (*Figure 4B*) (*Staub et al., 2004*). C-terminal GFP tagging of these *Tetrahymena* Top2 isoforms revealed distinct nuclear localization patterns: TTHERM_00456750p was expressed exclusively in the MAC and TTHERM_00825440p in MIC and MIC-derived pronuclei (*Figure 4C*). Hence, it is reasonable to assume that TTHERM_00825440p is the sole *Tetrahymena* functional homolog of mammalian Top2β to be exclusively expressed in spermatids (*Leduc et al., 2008*). Furthermore, the common ancestor of ciliates might have acquired multiple *TOP2* genes to manage its nuclear dualism. Hereafter, we will refer to TTHERM_00825440p as Top2G (Germline Top2) and TTHERM_00456750p as Top2S (Somatic Top2).

To investigate the post-meiotic function of Top2G, we made RNA interference (RNAi) strains to knock down gene expression (*Figure 5A*). Western blotting showed that Top2G-GFP, which was most abundant at post-meiosis (i.e. 4 hr and later after meiosis induction), was completely depleted by RNAi expression (*TOP2gi*; *Figure 5B*), demonstrating that *TOP2* knockdown was effective. Acetic orcein staining of chromosomes showed that Top2G depletion did not affect meiosis: four normal pronuclei were generated, as in the wild-type cross (*Figure 5C*). However, gametogenic mitosis failed in the *TOP2gi* cross: all pronuclei migrated to the posterior region of the cytoplasm, similar to unselected pronuclei in the wild-type (*Figure 5C*). These pronuclei had vanished by 10 hr, probably via autophagy (*Figure 5C*) (*Liu and Yao, 2012*), rendering about 70–80% of the *TOP2gi* crosses amicronucleate (*Figure 5D* and *Figure 5—source data 1*). Mating was eventually aborted in amicronucleate cells, without the formation of progeny nuclei (*Figure 5C*).

To see whether Top2G depletion affects γH2AX formation in the pronuclei, γH2AX was immunostained in the *TOP2gi* cross. Similar to the effect of Top2β chemical inhibition on animal spermiogenesis (*Laberge and Boissonneault, 2005*), loss of Top2G activity in *Tetrahymena* suppressed H2AX phosphorylation in the pronuclei (*Figure 5E,F* and *Figure 5—source data 2*). However, γH2AX formation at meiotic prophase was not affected (*Figure 5E*). This result suggests that Top2G may be required to form post-meiotic DNA lesions but not meiotic DSBs.

## Possible involvement of Spo11 in post-meiosis

Spo11 induces practically all meiotic DSBs needed for bivalent formation and genetic exchange by homologous recombination (*Keeney, 2008*). *Gouraud et al. (2013)* suggested that Spo11 has an additional function as a PM-DSB inducer based on microarray data (http://www.ncbi.nlm.nih.gov/geo/query/acc.cgi?acc=GSE2736) showing that *SPO11* transcription remains high in elongating spermatids in mice. Similarly, RT-PCR showed that *SPO11* is transcribed from meiotic prophase (~3 hr) until the post-meiotic stage (6 hr) in *Tetrahymena* (*Figure 6C*), suggesting that Spo11 functions in post-meiotic processes.

The meiotic role of *Tetrahymena* Spo11 in DSB formation and nuclear elongation was elucidated using somatic knockout (*ΔSPO11*) and knockdown (*SPO11i*) lines (*Mochizuki et al., 2008*; *Loidl and Mochizuki, 2009*; *Howard-Till et al., 2013*). Here, the post-meiotic phenotypes of these mutants were assessed by acetic orcein staining. Both mutants displayed indistinguishable phenotypes at meiotic prophase: neither DSBs nor γH2AX were formed in the meiotic MIC and nuclear elongation did not take place. Nevertheless, meiotic anaphases I and II were not affected, and four pronuclei were formed as in the wild-type crosses (*Figure 6A*). However, similar to the *TOP2gi* crosses (*Figure 5C*), the *ΔSPO11* mutant did not undergo gametogenic mitosis, and only amicronucleate single cells were present by 10 hr after meiosis induction (*Figure 6A,B* and *Figure 6—source data 1*). These results suggest a role for Spo11 not only in meiotic prophase but also at the post-meiotic stage. Interestingly, the post-meiotic defect was less severe in *SPO11i* crosses: progeny nuclei developed following gametogenic mitosis in 40–70% of these cells (*Figure 6A,B* and *Figure 6—source data 1*). Attenuation of the mutant phenotype suggests that *SPO11* expression recovers from RNAi-

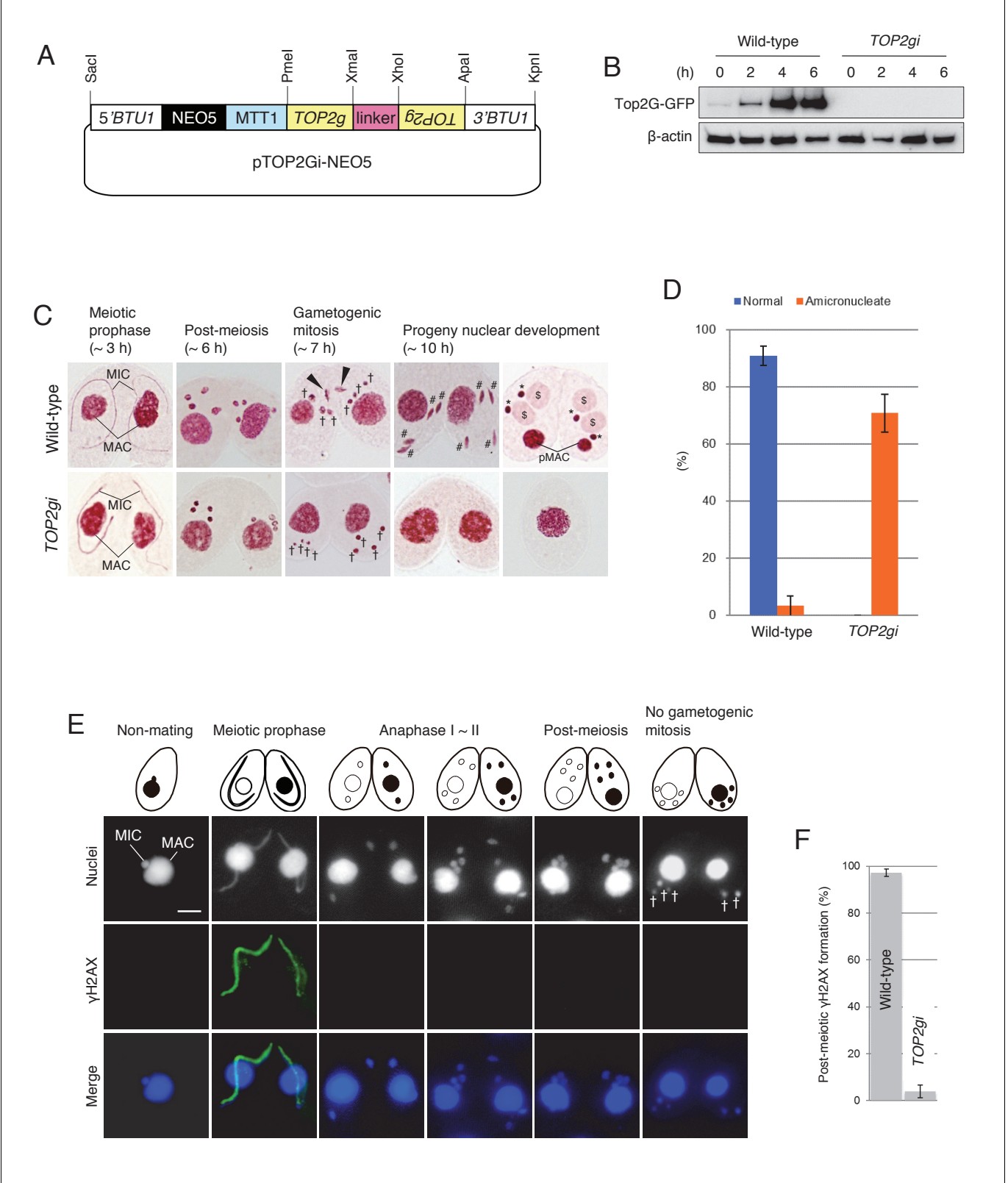

**Figure 5.** Role of Top2G in the post-meiotic stage. (**A**) Schematic representation of the pTOP2Gi-NEO5 knockdown vector used for Top2G RNAi. Hairpin RNA was produced under the control of the cadmium-inducible *MTT1* promoter. This construct was integrated into the *β*-tubulin (*BTU1*) genomic locus by homologous recombination. (**B**) Western blotting of TOP2G-GFP in the wild-type and *TOP2gi* crosses to assess RNAi efficiency. *β*-actin was the loading control. (**C**) Acetic orcein staining of the wild-type (top) and *TOP2gi* (bottom) crosses. Gametogenic mitosis (black arrowheads)

*Figure 5 continued on next page*

*Figure 5 continued*

took place in the wild-type crosses but was not observed in the *TOP2gi* crosses. Wild-type crosses underwent progeny nuclear development by 10 hr, whereas *TOP2gi* crosses became amicronucleate and did not produce progeny nuclei. MAC-macronucleus; MIC-micronucleus; †-unselected pronuclei; #-differentiating zygotic nuclei; $-progeny macronuclear anlagen; *-progeny micronuclei; pMAC-degrading parental macronuclei. Time (h) after mixing starved cells of different mating types is indicated. (D) Percentage of wild-type and *TOP2gi* cells with progeny nuclei and amicronucleate cells at 10 hr. Columns and error bars represent the means and standard deviations (p<0.01 as calculated by Tukey HSD test) of four measurements. See also **Figure 5–sourse data 1** for wild-type and *TOP2gi* crosses data. (E) Post-meiotic γH2AX formation is inhibited in *TOP2gi* crosses (compare with wild-type crosses in **Figure 1A**). Unselected pronuclei (†) assemble in the posterior region of the cytoplasm and are degraded. Scale bar denotes 10 μm. (F) Percentage of wild-type and *TOP2gi* post-meiotic cells with γH2AX formation in the pronuclei. Columns and error bars represent means and standard deviations (p<0.01, as calculated by Tukey's HSD test) of four measurements. See also **Figure 5–sourse data 2** for wild-type and *TOP2gi* crosses data.

The following source data is available for figure 5:

**Source data 1.** Normal development is significantly reduced in *TOP2gi* crosses.
**Source data 2.** Post-meiotic γH2AX formation is significantly reduced in *TOP2gi* crosses.

mediated knockdown after meiotic prophase. To measure Spo11 protein expression, we created C-terminally FZZ-tagged (**Lee and Collins, 2007**) Spo11-expressing strains. By Western blotting, Spo11-FZZ expression was first seen in wild-type crosses at meiotic prophase (2 hr), reached a maximum during anaphase I to II (~3–4 hr), and remained at that level until post-meiosis (6 hr; **Figure 6D**). In the presence of *SPO11* RNAi, Spo11-FZZ was not expressed during meiotic prophase (~2–3 hr) but was expressed at later stages (~4–6 hr), including the post-meiotic stage (**Figure 6D**). This result indicates that the RNAi interferes with *SPO11* expression at meiotic prophase, but its effect is gradually lost at later stages.

γH2AX immunostaining was performed to determine whether *SPO11* expression at the post-meiotic stage correlates with pronuclear H2AX phosphorylation. As demonstrated previously (**Mochizuki et al., 2008**; **Loidl and Mochizuki, 2009**; **Howard-Till et al., 2013**), neither the *ΔSPO11* nor *SPO11i* crosses expressed γH2AX in the meiotic MIC (**Figure 6E**). Moreover, none of the *ΔSPO11* crosses and only 40–70% of the *SPO11i* crosses showed γH2AX immunostaining in the pronuclei (**Figure 6E,F** and **Figure 6—source data 2**). These results suggest that Spo11 plays an additional role in inducing DNA lesions at the post-meiotic stage, together with Top2G (**Figure 5**).

## Direct evidence for post-meiotic DSB

Since γH2AX formation is an indirect marker of DSBs, we carried out pulsed-field gel electrophoresis (PFGE) to detect chromosome fragmentation, which is diagnostic of DSBs (**Lukaszewicz et al., 2013**). DNA was electrophoresed for 72 hr to separate fragment sizes of 0.01–5.7 Mb. Intact MIC chromosomes ($n = 5$, ~25.0–35.0 Mb) (**Hamilton et al., 2016**) do not enter the gel under these conditions, and gel staining showed only the MAC minichromosomes (**Figure 7A**). The smallest band represents ribosomal DNA (rDNA; **Figure 7A**, red arrow) of size ~0.02 Mb (**Løvlie et al., 1988**), which was used as the loading control.

To distinguish MIC chromosome fragments from the MAC minichromosomes, MIC DNA was detected by Southern blotting with a $^{32}$P-labeled Tlr sequence, which is specific to the MIC genome (**Wuitschick et al., 2002**). On Southern blots of mating cells, four different DNA patterns could be distinguished (**Figure 7B**). For control cells at t = 0 hr, no DNA complementary to the Tlr probe entered the gel, indicating the presence of intact MIC chromosomes. Mating cells at t = 2–4 hr displayed a prominent smear between 2.2 and 4.6 Mb (**Figure 7B**, blue box). This signal represents meiotic DSBs induced by Spo11, because it was absent in both the *ΔSPO11* and *SPO11i* crosses but present in *TOP2gi* crosses (**Figure 7B**). At the post-meiotic stage (t = 6–7 hr), a smear of shorter DNA fragments appeared in all cell lines (**Figure 7B**, green box). This DNA signal probably originates from the autophagic degradation of unselected pronuclei (**Liu and Yao, 2012**). In addition, a class of somewhat larger fragments was present at t = 6–7 hr in the wild-type and *SPO11i* crosses, but absent in the *TOP2gi* and *ΔSPO11* crosses (**Figure 7B**, red box). The absence of this subset of DNA fragments corresponds with the absence of pronuclear γH2AX formation in the *TOP2gi* and *ΔSPO11* crosses (**Figures 5E** and **6E**). Hence, these fragments are probably created by PM-DSBs.

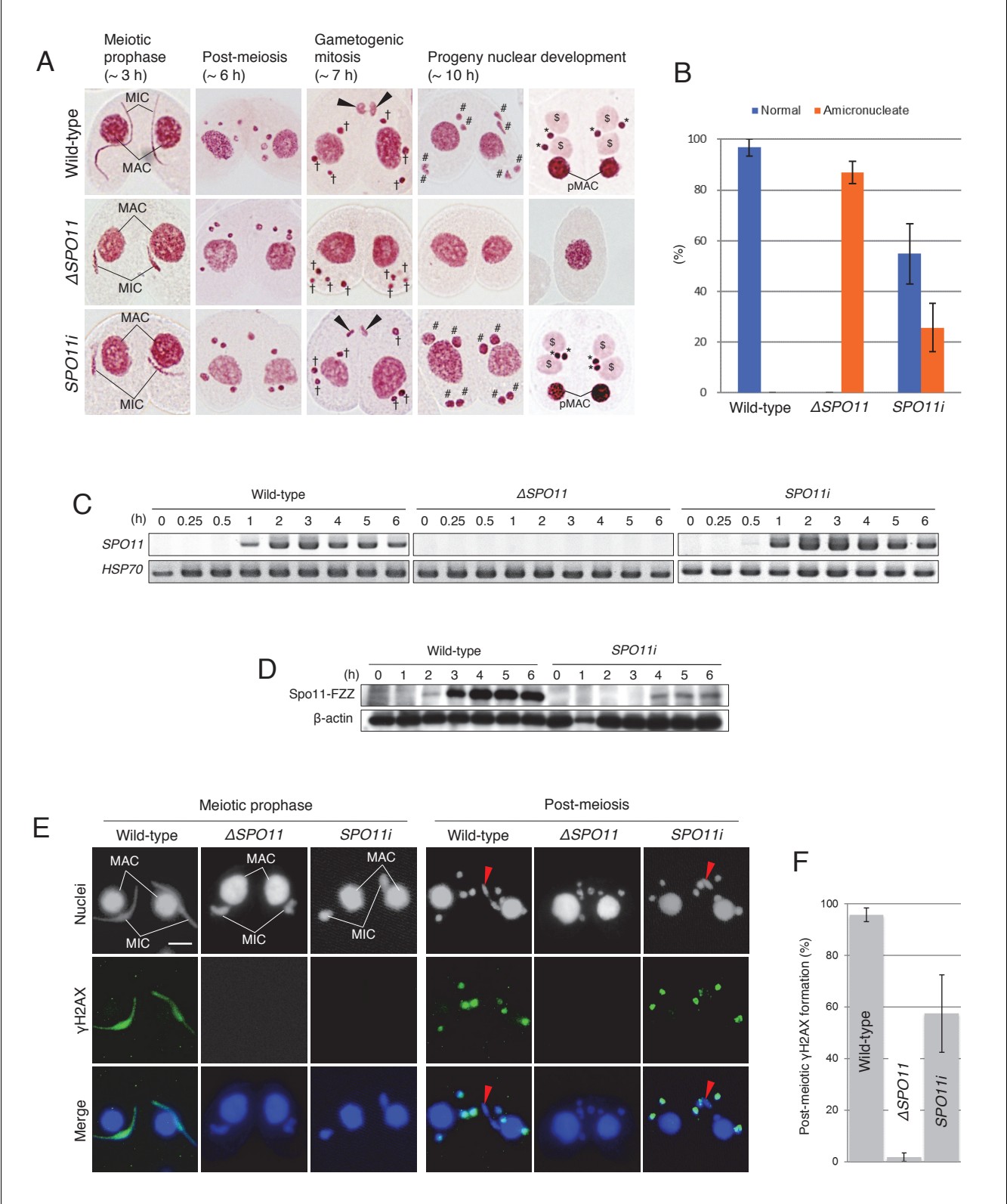

**Figure 6.** Spo11 is required for the correct execution of post-meiotic events. (**A**) Acetic orcein staining of wild-type (top), *ΔSPO11* (middle), and *SPO11i* (bottom) crosses. Although *ΔSPO11* crosses display the same post-meiotic phenotype as *TOP2gi* crosses (cells become amicronucleate and do not undergo gametogenesis; see *Figure 4C*), some *SPO11i* crosses behave in the same way as wild-type crosses by producing normal progeny nuclei via gametogenic mitosis (black arrowheads). MAC-macronucleus; MIC-micronucleus; †-unselected pronuclei; #-zygotic nuclei being differentiated; $-

*Figure 6 continued on next page*

*Figure 6 continued*

progeny macronuclear anlagen; *-progeny MICs; pMACs-degrading parental macronuclei. (B) Percentage of cells with progeny nuclei and amicronucleate cells at 10 hr. Columns and error bars represent means and standard deviations (p<0.01 as calculated by Tukey's HSD test) of four measurements. See also *Figure 6–sourse data 1* for data on wild-type, *ΔSPO11*, and *SPO11i* crosses. (C) RT-PCR quantitation of *SPO11* transcription in wild-type, *ΔSPO11*, and *SPO11i* crosses. *HSP70* was the loading control. Time (h) after mixing cells is indicated. (D) Western blotting of C-terminally FZZ-tagged Spo11 in wild-type and *SPO11i* crosses shows the impact of RNAi on SPO11-FZZ expression during the pre-zygotic period of mating. *β*-actin was the loading control. Time (h) after mixing cells is indicated. (E) SPO11 is involved in both meiotic and post-meiotic γH2AX formation. Although γH2AX foci are not seen at the meiotic or post-meiotic stages in *ΔSPO11* crosses, post-meiotic γH2AX is formed normally in a subset of *SPO11i* crosses in which Spo11 is expressed at the post-meiotic stage (see *Figure 5D*). Red arrowheads indicate the selected pronuclei in gametogenic mitosis from which γH2AX has been lost. Scale bar denotes 10 μm. (F) Percentage of post-meiotic cells with γH2AX in the pronuclei. Columns and error bars represent means and standard deviations (p<0.01 as calculated by Tukey's HSD test) of four measurements. See also *Figure 6—source data 2* for data on wild-type, *ΔSPO11*, and *SPO11i* crosses.

The following source data is available for figure 6:

**Source data 1.** Normal development is significantly reduced in *ΔSPO11* and *SPO11i* crosses.

**Source data 2.** Post-meiotic γH2AX formation is significantly reduced in *ΔSPO11* and *SPO11i* crosses.

Since autophagy of unselected pronuclei overlaps temporally with post-meiotic transformation of the selected pronucleus, it is still possible that the DNA smear seen at *t* = 6–7 in the wild-type and *SPO11i* crosses results entirely from DNA degradation in unselected pronuclei. To eliminate

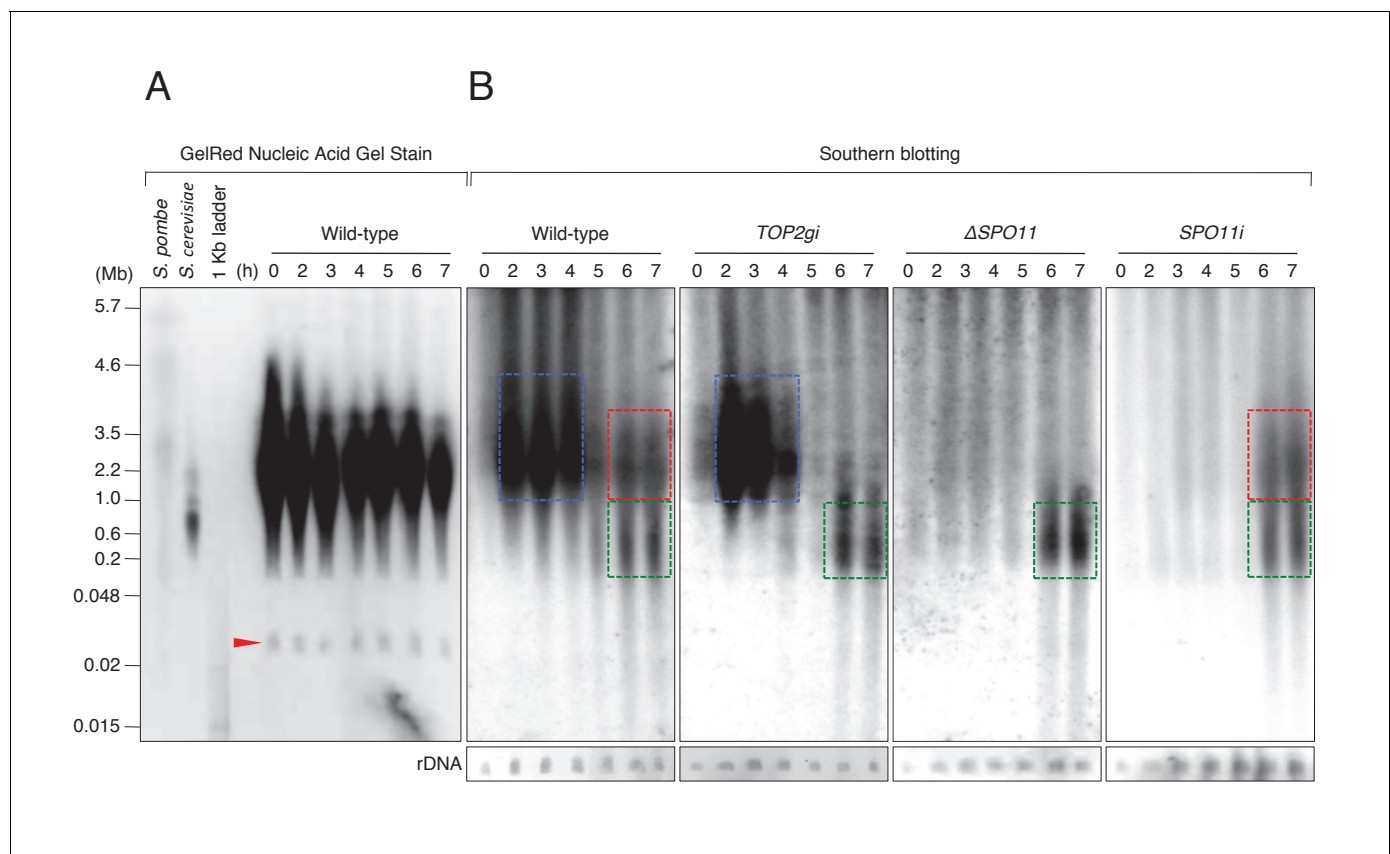

**Figure 7.** PFGE followed by Southern blotting provides direct evidence of post-meiotic DSBs. (A) Example of a pulsed-field gel with stained DNA. The red triangle indicates rDNA, the loading control. (B) Southern blotting using the Tlr element as a germline-specific probe. Smear signals represent DSB-generated chromosome fragments in the meiotic MIC (blue squares) and pronuclei (red and green squares), whereas intact micronuclear chromosomes (25.0–35.0 Mb) did not enter the gel. Time (h) after mixing cells is indicated.

background autophagic DNA fragments, we created somatic knockout strains for the autophagy-related 8 isoform 2 gene (*ATG8-2*), which encodes a central component of the *Tetrahymena* autophagosome (*Figure 8A*; *Liu and Yao, 2012*), using the co-Deletion technique (described in *Hayashi and Mochizuki, 2015*). Diagnostic PCR for the *ΔATG8-2* strains of different mating types showed that *ATG8-2* genomic loci were completely deleted (*Figure 8B*). The *ΔATG8-2* crosses underwent mating, including normal meiosis and γH2AX formation in the pronuclei (*Figure 8C*), but were defective in eliminating unselected pronuclei; therefore, post-meiotic cells contained extra nuclei (*Figure 8D*). PFGE detection of DNA fragments revealed normal meiotic DSBs plus a prominent post-meiotic (6–7 hr) DSB signal, which is unlikely to result from autophagy (*Figure 8E*). Next, we tested *TOP2gi::ΔATG8-2*, *ΔSPO11::ΔATG8-2*, and *SPO11i::ΔATG8-2* double mutants in the same assay. In the *TOP2gi::ΔATG8-2* mutant, meiotic DSB-dependent fragmentation was normal but post-meiotic fragmentation was virtually absent. In the *ΔSPO11::ΔATG8-2* mutant, meiotic DSB-dependent fragmentation was absent and post-meiotic fragmentation was strongly reduced. Finally, in the *SPO11i::ΔATG8-2* mutant, the meiotic signal was absent and the post-meiotic signal was partly retained (*Figure 8E*). The post-meiotic signal in this mutant was completely absent in a triple mutant (*TOP2gi::SPO11i::ΔATG8-2*; *Figure 8E*). These results show that programmed PM-DSBs are present in *Tetrahymena*, suggesting that PM-DSB formation requires both Top2g and Spo11.

## The selected pronucleus undergoes chromatin remodeling

DNA repair factors DNA-PKcs and Rad51 were expressed (*Figure 3—figure supplements 1* and *2*), and γH2AX immunoreactivity was lost in the selected pronucleus prior to gametogenic mitosis (*Figure 3C,D*), suggesting that DNA repair had been accomplished. However, since γH2AX dephosphorylation is only an indirect mark of repair (*Chowdhury et al., 2005*; *Keogh et al., 2006*), we looked for additional evidence that repair occurs in the selected pronucleus. Canonical histones around DNA lesions are often removed by chromatin remodelers to allow access by the repair machinery (*Altaf et al., 2007*; *Osley et al., 2007*). Newly synthesized histone H3 acetylated at lysine 56 (H3K56ac) is then deposited onto the repaired DNA (*Chen et al., 2008*; *Shi and Oberdoerffer, 2012*). Therefore, H3K56ac immunostaining was performed in mating wild-type cells to confirm that this process occurs in *Tetrahymena*. Consistent with their euchromatic state, wild-type MACs were positive for H3K56ac staining (*Figure 9A*). In contrast, owing to their heterochromatic state, MICs did not undergo acetylation in either vegetative or meiotic cells (*Figure 9A*) (*Garg et al., 2013*). However, we found that the selected pronucleus became H3K56 acetylated as it started to undergo gametogenic mitosis, while the three unselected pronuclei did not become acetylated and were degraded (*Figure 9A*). Double immunostaining for H3K56ac and γH2AX confirmed that acetylation and dephosphorylation occurred in the same pronucleus (*Figure 9B*), indicating that the selected pronucleus specifically undergoes DNA repair and histone H3 transfer. In the *ΔATG8-2* crosses, H3K56 acetylation and H2AX dephosphorylation occurred in the selected pronucleus just as in the wild type (*Figure 9C*). The remaining unselected pronuclei also showed a decline in γH2AX staining after post-meiotic mitosis; however, neither H3K56 acetylation nor gametogenic mitosis occurred in these nuclei (*Figure 9C*). These results indicate that DNA repair and histone H3 modification are not merely the consequence of a nucleus escaping autophagy.

H3K56 acetylation upon DNA repair is catalyzed by the histone acetyltransferase Rtt109 (*Chen et al., 2008*), and the histone chaperone anti-silencing factor 1 (Asf1) is essential for Rtt109 stimulation and H3K56ac assembly at repaired DNA lesions (*Recht et al., 2006*; *Tsubota et al., 2007*; *Berndsen et al., 2008*; *Shi and Oberdoerffer, 2012*). *Tetrahymena* Asf1 is expressed in both the MAC and MIC during vegetative growth (*Garg et al., 2013*). We used a GFP-tagged Asf1-expressing strain to investigate changes in protein localization during mating. Asf1 was retained in the elongating MIC at meiotic prophase but lost during anaphase I and II (*Figure 10A*). After completion of meiosis, Asf1 reappeared only in the pronucleus selected to undergo gametogenic mitosis (*Figure 10A*). Double immunostaining demonstrated Asf1-GFP and H3K56ac colocalization in the selected pronucleus (*Figure 10B*), strongly suggesting that histone chaperone-mediated nucleosome assembly precedes gametogenic mitosis. Importantly, H3K56 acetylation potentially mediated by Asf1 was only induced in the selected pronucleus in response to PM-DSBs, as neither Asf1-GFP nor H3K56ac was observed when PM-DSBs were suppressed by *TOP2g* RNAi or *SPO11* deletion (*Figure 10B,C* and *Figure 10—source data 1*).

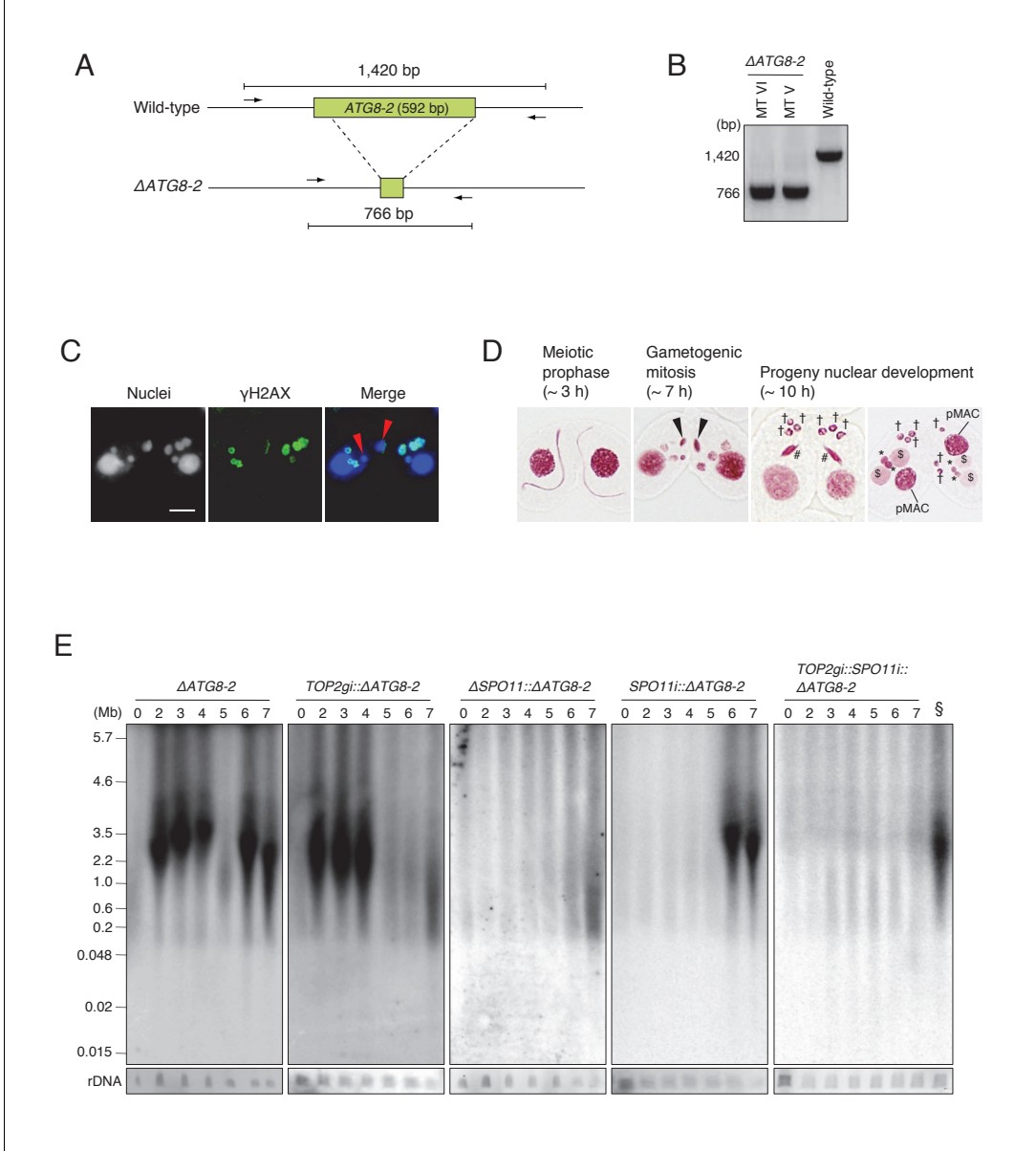

**Figure 8.** *ATG8-2* gene knockout eliminates background DNA fragments resulting from autophagic degradation of unselected pronuclei. (**A**) Schematic representation of *ATG8-2* knockout strains by co-Deletion (as described in *Hayashi and Mochizuki, 2015*), confirmed by diagnostic PCR. The PCR primer set is indicated by arrows. (**B**) Only fragments from the deleted *ATG8-2* genomic locus were amplified in *ΔATG8-2* strains of two different mating types, while only fragments from the intact *ATG8-2* genomic locus were amplified from the wild-type strain. (**C**) Post-meiotic γH2AX formation was not affected in the *ΔATG8-2* crosses: gametogenic mitosis (red arrowheads) took place as in wild-type crosses (see *Figure 2C/c/C'*). Scale bar denotes 10 μm. (**D**) Acetic orcein staining of the *ΔATG8-2* crosses shows that unselected pronuclei (†) are retained beyond gametogenic mitosis (black arrowheads) owing to attenuated autophagy. #-differentiating zygotic nuclei; $-progeny macronuclear anlagen; *-progeny MICs; pMACs-degrading parental macronuclei. (**E**) Southern hybridization of *ΔATG8-2*, *TOP2gi::ΔATG8-2*, *ΔSPO11::ΔATG8-2*, *SPO11i::ΔATG8-2*, and *TOP2gi::SPO11i::ΔATG8-2* strains. The *SPO11i::ΔATG8-2* strain (in which both meiotic and autophagic DSBs are eliminated) retains only the signal resulting from PM-DSBs. rDNA was the loading control. Time (h) after mixing cells is indicated. § indicates a *SPO11i::ΔATG8-2* 7 hr sample as the as the positive control for the *ΔATG8-2:: TOP2gi::SPO11i* blot. See also *Figure 8—figure supplement 1*.

The following figure supplement is available for figure 8:

**Figure supplement 1.** 5-Ethynyl-2'-deoxyuridine (EdU) incorporation assay.

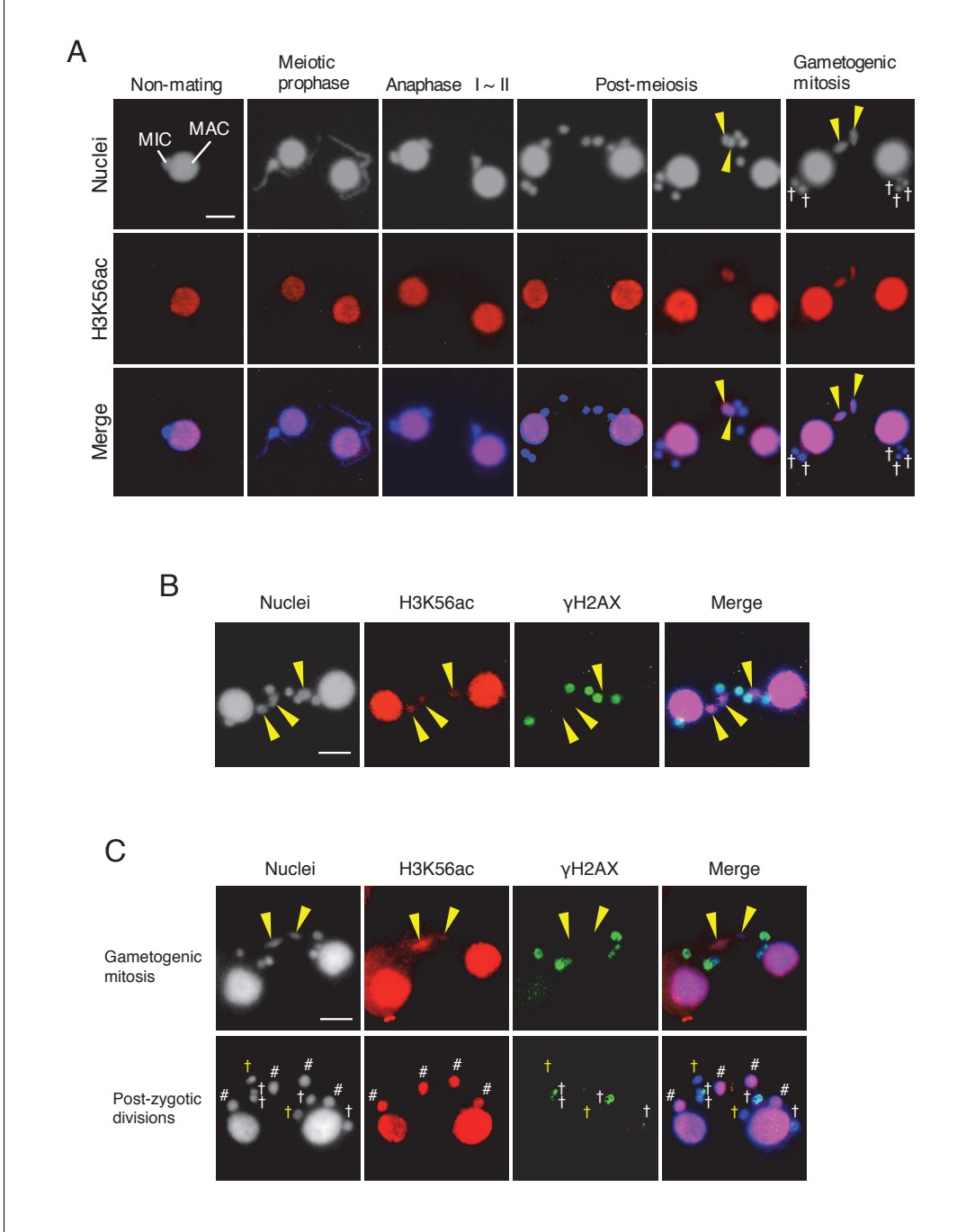

**Figure 9.** Histone H3 acetylation at lysine 56 (H3K56ac) is concomitant with H2AX dephosphorylation in the selected pronucleus. (**A**) H3K56ac in the pre-zygotic period in the wild-type. The MAC was always H3K56ac positive, while the MIC only became H3K56ac positive as the pronucleus underwent gametogenic mitosis (yellow arrowheads). MAC-macronucleus; MIC-micronucleus, †-unselected pronuclei. (**B**) H3K56ac formation and H2AX dephosphorylation are coincident at the post-meiotic stage. The selected pronuclei undergoing gametogenic mitosis (yellow arrowheads) are H3K56ac-positive but γH2AX-negative. (**C**) H3K56 acetylation does not occur in persisting unselected nuclei in the *ΔATG8-2* mutant, whereas γH2AX is reduced in some of the persistent unselected pronuclei (yellow †) in the post-zygotic period. In the selected pronucleus, H3K56ac formation and H2AX dephosphorylation are normal (yellow arrowheads). White †-unselected pronuclei with persistent γH2AX; #-differentiating zygotic nuclei. Scale bars denote 10 μm. See also *Figure 9—figure supplement 1*.

The following figure supplement is available for figure 9:

**Figure supplement 1.** Histone H3 acetylation at sites other than H3K56 is coincident with H2AX dephosphorylation in the selected pronucleus.

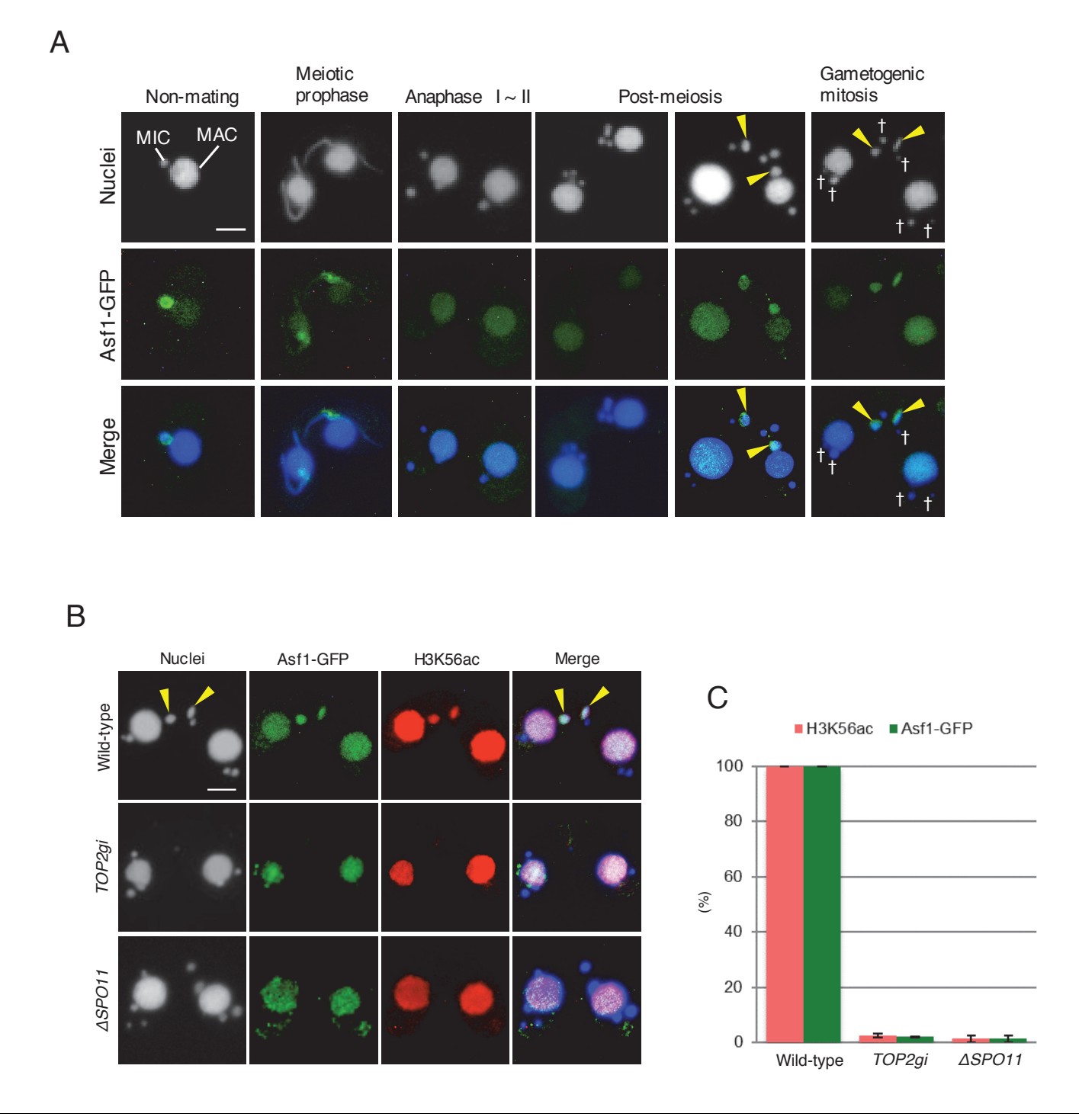

**Figure 10.** Histone chaperone Asf1 is specifically expressed in the selected pronucleus. (**A**) Subcellular localization of C-terminally GFP-tagged Asf1 in the pre-zygotic mating stage in wild-type cells. Asf1 disappears from the MIC after meiotic prophase and reappears in the selected pronuclei undergoing gametogenic mitosis (yellow arrowheads). MAC-macronucleus; MIC-micronucleus; †-unselected pronuclei. (**B**) Asf1 and H3K56ac co-expression in the selected pronuclei is PM-DSB dependent. In wild-type crosses (top), selected pronuclei undergoing gametogenic mitosis (yellow arrowheads) are positive for both ASF1-GFP and H3K56ac. In contrast, both signals are absent in the *TOP2gi* (middle) and *ΔSPO11* crosses (bottom), which do not form PM-DSBs (see *Figure 6B* and *Figure 7E*). Scale bars denote 10 μm. (**C**) Percentage of post-meiotic cells with H3K56ac and Asf1-GFP in the selected pronucleus. Columns and error bars represent the means and standard deviations (p<0.01 as calculated by Tukey's HSD test) of four measurements. See also *Figure 10—source data 1* for wild-type, *TOP2gi*, and *ΔSPO11* crosses.

*Figure 10 continued on next page*

*Figure 10 continued*

The following source data is available for figure 10:

**Source data 1.** Appearance of H3K56ac and Asf1-GFP is significantly reduced in *TOP2gi* and *ΔSPO11* crosses.

## Discussion

### PM-DSBs are deliberately induced

DSBs caused by external and internal agents are dangerous lesions, so it is important to understand why they should be deliberately induced. In fact, self-inflicted DSBs are essential for a wide range of processes, such as TOP2-dependent DNA disentangling (*Champoux, 2001*), V(D)J recombination (*Stavnezer et al., 2008*), chromatin diminution (*Wang and Davis, 2014*), *Saccharomyces cerevisiae* mating-type switching (*Haber, 2012*), antigenic variation in *Trypanosoma* (*McCulloch et al., 2015*), and, most prominently, meiosis (*Keeney et al., 1997*). They may also be essential for chromatin remodeling.

In metazoans, the products of male meiosis undergo chromatin remodeling so that genetic material can be densely packed into small sperm (*Miller et al., 2010*; *Rathke et al., 2014*). Accordingly, it has been claimed that PM-DSBs are neccessary to support chromatin remodeling during spermiogenesis (*Marcon and Boissonneault, 2004*; *Leduc et al., 2008*; *Rathke et al., 2014*). After fertilization, male chromatin unpacking and epigenetic reprogramming (*McLay and Clarke, 2003*; *Seisenberger et al., 2013*) are also accompanied by DNA repair (*Schuermann et al., 2016*). Little is known, however, about post-meiotic chromatin changes in other contexts. Here, we show that PM-DSBs are induced in the pronuclei of *Tetrahymena* and that the Top2-related proteins, Top2G and Spo11, are essential for inducing these DSBs. Our data suggest that PM-DSBs play an indispensable role in producing gametic nuclei by converting their chromatin from a heterochromatic to a euchromatic state.

Top2$\beta$, which is exclusively expressed in elongating spermatids, is a candidate for inducing transient DNA lesions during animal spermiogenesis (*Leduc et al., 2008*). In the green algae *Chara vulgaris*, the topoisomerase inhibitor etoposide interferes with post-meiotic γH2AX formation and spermatogenesis (*Agnieszka and Wojtczak, 2014*). This observation was interpreted as demonstrating a need for transient Top2-induced DSBs to exchange nucleoproteins. By depleting Top2G (the *Tetrahymena* ortholog of mammalian Top2$\beta$) by RNAi (*Figure 4A,B*), we provide the first direct evidence that PM-DSB formation is Top2 dependent (*Figure 8E*). Moreover, we demonstrate that Spo11 is expressed after meiosis (*Figure 6C,D*) and is required for inducing PM-DSBs (*Figure 8E*).

In principle, either Top2g or Spo11 should be able to induce DSBs; it is therefore surprising that both proteins are required for PM-DSB formation. It is unclear whether these two proteins cooperate within a complex and which of them is directly involved in DNA cleavage. In reciprocal co-immuno-precipitation experiments with Spo11-FZZ as the bait and Top2g-GFP as the prey (and vice versa), we could not confirm that these two proteins interact.

### Possible PM-DSB repair mechanisms

As spermatids are haploid $G_1$ cells, in which recombinational repair is impossible, NHEJ was considered the only possible repair pathway for putative mouse PM-DSBs (*Leduc et al., 2008*). The discovery of Ku70 and Mre11, both core NHEJ factors (*Chiruvella et al., 2013*), in the extracts of mouse and grasshopper spermatids (*Goedecke et al., 1999*; *Hamer et al., 2003*; *Cabrero et al., 2007*) supports this hypothesis. Since NHEJ is inherently error prone, PM-DSBs are considered substrates for male-driven de novo mutations (*Grégoire et al., 2013*).

In contrast to other organisms, *Tetrahymena* pronuclei are in the $G_2$ phase of the cell cycle (*Cole and Sugai, 2012*). 5-Ethynyl-2′-deoxyuridine (EdU) incorporation indicated that DNA synthesis is not impeded in either the *TOP2gi* or *ΔSPO11* crosses (*Figure 8—figure supplement 1*), strongly suggesting that PM-DSBs are formed after DNA synthesis. Therefore, it is possible that *Tetrahymena* repairs PM-DSBs by recombinational repair using the sister chromatid as the template. Indeed, we found the recombination protein Rad51 in the selected pronucleus (*Figure 3—figure supplement 2*). However, its function in PM-DSB repair is difficult to confirm experimentally because its depletion

arrests meiotic prophase (*Howard-Till et al., 2011*). On the other hand, we also found DNA-PKcs in the selected pronucleus (*Figure 3—figure supplement 1*), suggesting that NHEJ may also be involved in the repair of *Tetrahymena* PM-DSBs. Based on these data, it is unclear whether *Tetrahymena* PM-DSBs are repaired by NHEJ, sister-dependent recombinational repair, or both of these pathways.

## Post-meiotic DSBs trigger chromatin remodeling

In *Tetrahymena*, Asf1 appears in the selected pronucleus in a DSB-dependent manner (*Figure 10B*). This conserved histone H3 chaperone is involved in histone H3 K56 acetylation and H3K56ac transfer onto nascent DNA, which leads to euchromatin formation (*Chen et al., 2008*). In humans, disassembly of histone H3 during DNA repair is a local event, and NHEJ and recombinational repair remove histones within regions of about 0.75 kb and 7 kb from the break sites, respectively (*Goldstein et al., 2013*; *Li and Tyler, 2016*). PFGE showed that in *Tetrahymena* the size of PM-DSB-dependent DNA fragments ranges from ~1.0 to 4.6 Mb (*Figure 8E*). If we assume an average fragment size of 2.2 Mb, we estimate that ~140 DSBs are distributed across the two 157 Mb genomes of haploid pronuclei in $G_2$. If the influence of PM-DSB on chromatin remodeling is has a similar range in *Tetrahymena* as in humans, then histone H3 should be removed from only ~1 Mb of the entire genome. However, it is possible that PM-DSBs trigger global euchromatin formation via as yet unidentified chromatin remodelers. Alternatively, local histone replacement may be sufficient to proceed to gametogenic mitosis.

Garg et al. (2013) found that *Tetrahymena* Asf1 interacts with an importin β isoform and another protein similar to human nuclear autoantigenic sperm protein (NASP), both of which are involved in histone transport pathways (*Jäkel et al., 1999*; *Mühlhäusser et al., 2001*; *Bowman et al., 2016*). Garg et al. (2013) identified another Asf1-interacting partner protein containing a BRCA1 C-terminal (BRCT) domain, which is found predominantly in proteins involved in cell cycle checkpoint functions that respond to DNA damage (*Bork et al., 1997*; *Yu and Chen, 2004*). Our experiments showed that the selected pronucleus also undergoes acetylation at sites other than at H3K56; for example, H3K18ac and H3K27ac (*Figure 9—figure supplement 1*) are strongly enriched in euchromatin (*Wang et al., 2008*; *Tie et al., 2009*). Further, a protein containing a high mobility group (HMG) box domain, which decreases the compactness of the chromatin fiber (*Agresti and Bianchi, 2003*; *Catez et al., 2004*), is abundantly expressed in the selected pronucleus (*Xu et al., 2013*). Together, these findings suggest that repair of programmed PM-DSBs in the haploid germline allows changes in the epigenetic landscape required to produce mature gametes (*Figure 11*).

In mammalian post-meiotic development, the male genome undergoes several waves of epigenetic modification. The first is chromatin remodeling, which facilitates sperm packaging by substituting the histone-based nucleosome structure with a protamine-based structure (*Rathke et al., 2014*). This process is believed to be DSB dependent (*Marcon and Boissonneault, 2004*; *Rathke et al., 2007*, *2014*). The second is histone replacement of protamines upon fertilization (*Rousseaux et al., 2008*). The third is erasure of parental epigenome marks in the zygote. During this process, methylated cytosines are replaced by unmodified cytosines via base excision repair (BER) (*Wu and Zhang, 2014*). Interestingly, when BER is compromised, γH2AX foci are formed, suggesting that DSBs are induced, which are repaired after $G_1$ (*Wossidlo et al., 2010*; *Ladstätter and Tachibana-Konwalski, 2016*). Thus, while mammals require karyogamy to induce the epigenetic modifications necessary for embryonic developmental programming (*Zhou and Dean, 2015*), *Tetrahymena* gametic nuclei can differentiate into MACs and MICs without karyogamy (*Fukuda et al., 2015*). Therefore, chromatin remodeling in *Tetrahymena* pronuclei might resemble zygote reprogramming rather than mammalian male pronuclear chromatin remodeling, in that the differentiated germline pronucleus is re-set to a dedifferentiated progenitor of new somatic and germline nuclei. Since dedifferentiated human cells such as cancer and embryonic stem cells display more H3K56ac compared with differentiated somatic cells (*Das et al., 2009*), it is possible that enhanced histone acetylation in *Tetrahymena* similarly promotes nuclear dedifferentiation via a similar mechanism.

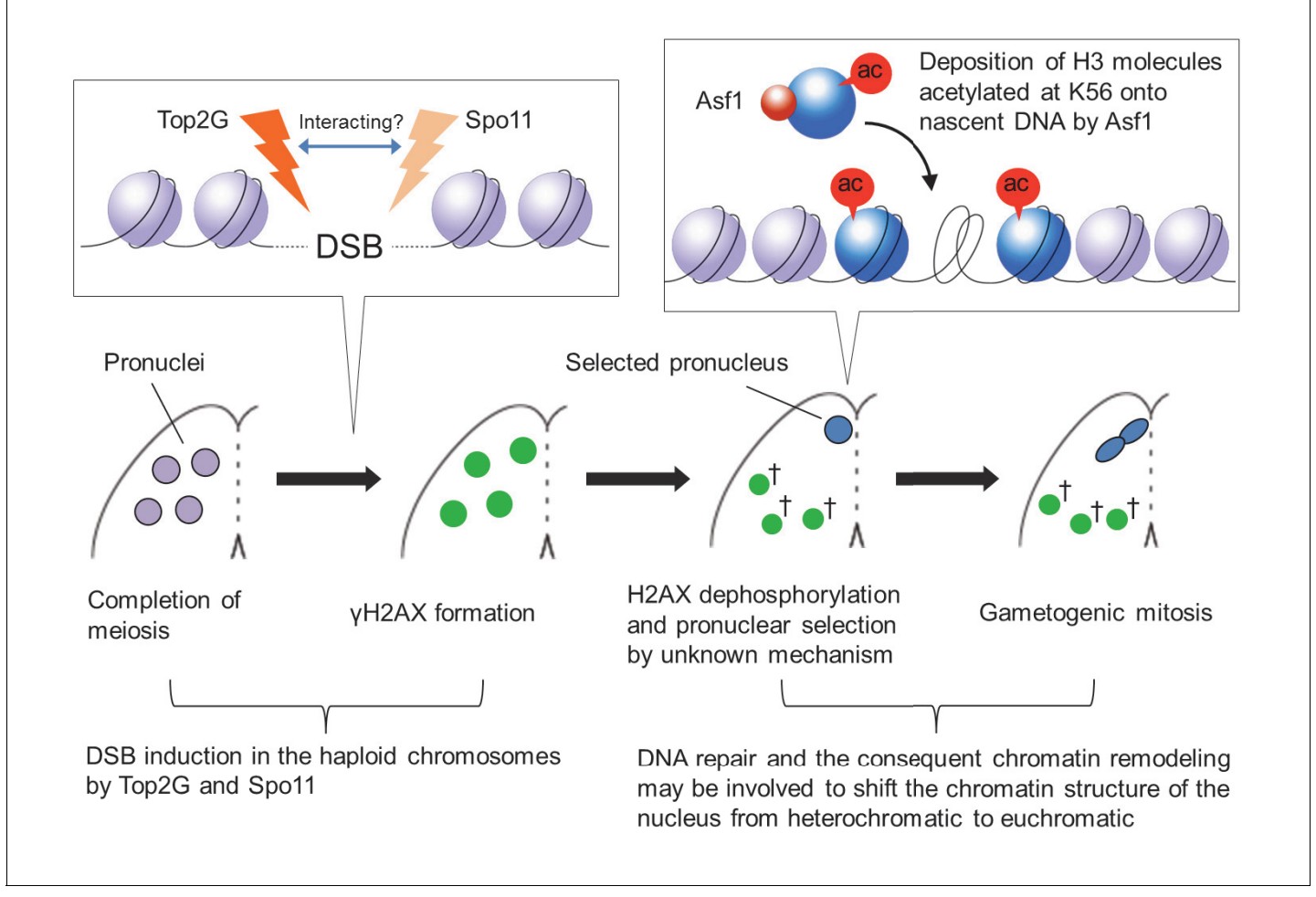

**Figure 11.** Model of post-meiotic events in *Tetrahymena*. After completion of meiosis, PM-DSBs induced in pronuclei haploid chromosomes by Top2G and Spo11 trigger H2AX phosphorylation. H2AX dephosphorylation, probably indicating DNA repair, takes place together with the Asf1 localization and H3K56ac formation in the pronucleus that is selected for gametogenic mitosis via an unknown mechanism. In the selected pronucleus, the chromatin structure changes from heterochromatic to euchromatic prior to gametogenic mitosis. The unselected pronuclei (†) retaining γH2AX are eventually eliminated via autophagy.

## Materials and methods

### Culture methods and the induction of cell mating (conjugation)

Wild-type *Tetrahymena* strains CU428.2 (mating type VII, RRID:TSC_SD00178), B2086 (mating type II, RRID:TSC_SD01627), and SB210 (mating type VI, RRID:TSC_SD00703) were obtained from the *Tetrahymena* Stock Center, Cornell University (https://tetrahymena.vet.cornell.edu/). *ΔSPO11* (*Mochizuki et al., 2008*), *SPO11i* (*Howard-Till et al., 2013*), and Asf1-GFP (*Garg et al., 2013*) strains were constructed previously. Cells were grown at 30°C in SPP medium containing 1% proteose peptone (Becton Dickinson, Sparks, MD, USA), 0.1% yeast extract (Becton Dickinson), 0.2% glucose (Sigma-Aldrich, St. Louis, MO), and 0.003% EDTA-Fe (Sigma-Aldrich). To make them competent for mating, cells at mid-log phase (approximately $10^6$/ml) were washed with 10 mM Tris-HCl (pH 7.4), resuspended in 10 mM Tris-HCl (pH 7.4), and starved at 30°C for ~18 hr to starve. To induce mating, equal numbers of cells of two different mating types were mixed together and incubated at 30°C.

## Indirect immunofluorescence

Cells were fixed with 0.34% Schaudinn's fixative (2:1 ration of saturated HgCl$_2$ [Sigma-Aldrich]: ethanol) and membrane-permeabilized with cold methanol on ice for 10 min. Cells were then spread onto slides coated with poly-l-lysine (Sigma-Aldrich) and air dried. After rehydration in PBS (4.3 mM Na$_2$HPO$_4$, 1.47 mM KH$_2$PO$_4$, 137 mM NaCl, 2.7 mM KCl, pH 7.4), cells were incubated for 2 hr at room temperature with primary antibody: anti-γH2AX (1:500; BioLegend, San Diego, CA, RRID:AB_315794), anti-GFP (1:500; mouse monoclonal; Sigma-Aldrich, RRID:AB_390913), or anti-H3K56 (1:500; rabbit polyclonal; Active Motif, Carlsbad, CA, RRID:AB_2661786) antibody. After washing with PBS, cells were incubated with FITC-labeled goat anti-mouse (1:500; Merck Millipore, Temecula, CA, RRID:AB_92634) or Rhodamine-labeled goat anti-rabbit (1:2000; Merck Millipore, RRID:AB_90296) secondary antibody for 1 hr at room temperature in the dark. After washing with PBS, cells were stained with 1 µg/µl 4′,6-diamidino-2-phenylindole (DAPI; Sigma-Aldrich) and observed under fluorescence microscopy.

## Western blotting

Cells were fixed with 10% (w/v) trichloroacetic acid (TCA; Sigma-Aldrich) to prevent proteolysis and incubated on ice for 30 min. After removal of TCA by centrifugation at 9000 $g$ for 1 min, cell pellets were lysed in PAGE sample buffer (2% SDS ([Sigma-Aldrich], 2.5% 2-mercaptoethanol [Sigma-Aldrich], 10% glycerol [Sigma-Aldrich], and 50 mM Tris-HCl, pH 6.8) and boiled at 98°C for 2 min; 10 µg total protein was loaded into each lane of a 12% polyacrylamide-SDS gel, separated by SDS-PAGE, and transferred onto polyvinylidene fluoride membrane (Merck Millipore). Membranes were washed in blocking buffer (5% dry skimmed milk powder [Sigma-Aldrich] in PBS) and incubated overnight at 4°C with anti-γH2AX (1:1000), anti-FLAG antibody (1:5000; Sigma-Aldrich, RRID:AB_259529), or anti-$\beta$-actin antibody (1:1000; GenScript, Piscataway, NJ, RRID:AB_914102). After washing in PBS-T (0.05% Tween 20 [Sigma-Aldrich] in PBS), membranes were incubated in PBS-T containing horseradish peroxidase-conjugated goat anti-mouse IgG antibody (1:5000; Thermo Fisher Scientific, Waltham, MA, RRID:AB_228307) for 1 hr at room temperature. Membranes were washed with PBS-T and developed using Clarity Western ECL (Bio-Rad, Hercules, CA).

## Acetic orcein staining

Cell suspension (30 µl) was pipetted onto a glass slide and air dried. The glass slide was then fixed in 3:1 methanol: acetic acid for 5 min, incubated in 5 N HCl for 5 min to degrade RNA, and then rinsed in distilled water for 10 s. Acetic orcein solution (5% orcein powder [Sigma-Aldrich] dissolved in 4.5:5.5 acetic acid: distilled water) was applied to the sample and stained nuclei were observed under light microscopy.

## Phylogenetic analysis

 *Tetrahymena* orthologs of mouse Top2α and Top2$\beta$ proteins were identified by their gene descriptions in the *Tetrahymena* genome database (http://www.ciliate.org). Orthologous protein sequences in other organisms were obtained from GenBank or dedicated databases for each species by performing BLASTp searches against mouse Top2α and Top2$\beta$. Complete amino acid sequences were used for multiple alignments with Clustal Omega (ver. 1.1.0) (*Sievers et al., 2011*). All gap regions appearing after alignment were eliminated from the sequences, and a phylogenetic tree was constructed for the resulting 996 amino acids Treefinder (*Jobb et al., 2004*) equipped with Aminosan software (*Tanabe, 2011*), which provided LG+I+G+F as the best evolutionary model for this data set. The phylogenetic tree was finally constructed using the maximum likelihood method in RAxML (Ver. 7.3.0) (*Stamatakis, 2006*). Confidence in the phylogeny was estimated using the bootstrap method in 100 replications.

## Construction of C-terminal epitope-tagged vectors

Approximately 1 kb of the ORFs (5′) and downstream UTRs (3′) of the *TOP2s*, *TOP2g*, and *SPO11* genomic loci were amplified from SB210 genomic DNA using PrimeSTAR Max DNA Polymerase (Takara, Kusatsu, Japan) and the following primers: *TOP2s* 5′ forward – AGTCGAGCTCACGCTAAG-GAGCAGACCTCG, reverse – AGTCGGATCCGAAATAGCATTCATCCGATGATTC; *TOP2s* 3′ forward – AGTCCTCGAGCATGCATTCATTCAATCAATCAATC, reverse – AGTCGGTACCGGTCTTGGCAA

TTAACTCTCTCAC; *TOP2g* 5′ forward – AGTCGAGCTCCAGGTAAAGGGTTTACATAGAATG, reverse – AGTCGGATCCATCATCCTCATCCTCATCAAATAA; *TOP2g* 3′ forward – AGTCCTCGAGA-CAGTGATGTCAGAATGTTAAATC, reverse – AGTCGGTACCCTTAAAGGCAGAAAATTAAGAGGT; *SPO11* 5′ forward – AGTCGAGCTCGATTACTGGGAAAGGGTA, reverse – AGTCGGATCCTAAATATTTGTTTGATTAGATTTTA; and *SPO11* 3′ forward – AGTCCTCGAGTAATTTCTTATTTTCTTTTTTGCT, reverse – AGTCGGTACCAATTTCTTCCATACAAAAAGCATCA). The 5′ ORF sequences do not contain a stop codon. Amplified PCR products were purified with a PCR Clean-up kit (Promega, Madison, WI), then 5′ sequences were digested with SacI plus BamHI (New England BioLabs, Ipswich, MA) and 3′ sequences with XhoI plus KpnI (New England BioLabs). These fragments were sequentially integrated into the backbone vector pEGFP-NEO4 or pFZZ-NEO4 (*Kataoka et al., 2010*) using T4 DNA ligase (New England BioLabs). The resulting vectors (pTOP2S-EGFP-NEO4, pTOP2G-EGFP-NEO4, and pSPO11-FZZ-NEO4) were linearized with SacI plus KpnI before biolistic transformation into *Tetrahymena* (*Cassidy-Hanley et al., 1997*).

### *TOP2g* RNAi vector construction and gene knockdown

The backbone plasmid pBNMB1-EGFP (a gift from Dr Kazufumi Mochizuki, Institute of Human Genetics, Montpellier, France) contains the *MTT1* cadmium-inducible promoter, a NEO5 drug-resistant marker, and the 5′ and 3′ portions of the *BTU1* genomic locus for homologous recombination. Target sequences used in hairpin constructs (approximately 700 b of the *TOP2g* ORF) were amplified from SB210 genomic DNA with PrimeSTAR Max DNA Polymerase (TaKaRa) using the following primers: *TOP2gi* 5′ forward – AGTCGTTTAAACCAGGTAAAGGGTTTACATAGAATGG, reverse – AGTCCCCGGGATTGCTCTTAGAAGGCATCATAACA; and *TOP2gi* 3′ forward – AGTCCTCGAGATTGCTCTTAGAAGGCATCATAACA, reverse – AGTCGGGCCCCAGGTAAAGGGTTTACATAGAATGG. Amplified forward and reverse target fragments were cloned into the PmeI-XmaI and XhoI-ApaI sites, respectively, of the pREC8hpCYH vector (a gift from Dr Rachel Howard-Till, University of Vienna, Austria) (*Howard-Till et al., 2013*) to create the hairpin cassette. The GFP cassette of the pBNMB1-EGFP plasmid was removed and replaced with the hairpin cassette digested with PmeI plus ApaI (New England BioLabs). The resulting vector (pTOP2Gi-NEO5) was linearized with SacI-KpnI before biolistic transformation into *Tetrahymena*. RNAi was induced in cells carrying the hairpin construct by adding 0.075 µg/ml CdCl$_2$ (Sigma-Aldrich) to promote dsRNA expression from the *MTT1* promoter.

### *ATG8-2* gene disruption

Approximately 0.6 kb of the *ATG8-2* ORF was amplified from SB210 genomic DNA using PrimeSTAR Max DNA Polymerase (TaKaRa) and the following primers: *ATG8-2* coDel forward – CTTTATTGTTATCATCTTATGACCGCGGACGCTCAAAATTATAAACCCTTC, reverse – CTCATCAAGTTGTAATGCTAAAATGCGCAAACACTACTGCATTTTCGCTAA. The amplified fragment was integrated into the backbone vector pMcoDel (*Hayashi and Mochizuki, 2015*) using Gibson Assembly Master Mix (New England BioLabs). The resulting vector (pMcoDel-ATG8-2) was used for biolistic transformation without linearization. Deletion of the target locus from the MAC was confirmed using a diagnostic primer set: Check *ATG8-2* forward – GAATAGAAAGTGCATCTCCTGATC, reverse – CTGGCAAACAAGAAGCACATTG.

### Replacement of paromomycin resistance markers with a puromycin resistant marker

To transfect the *SPO11* disruption (pΔSPO11-NEO4; *Mochizuki et al., 2008*), pTOP2Gi-NEO5, and C-terminally GFP-tagged Asf1 expression (pASF1-GFP-NEO2; *Garg et al., 2013*) vectors into paromomycin-resistant mutant strains, we replaced the NEO-based drug resistance markers in the vectors with a puromycin resistance marker (PAC; *Iwamoto et al., 2014*). NEO cassettes were removed from the vectors by digesting with SalI plus XmaI (New England BioLabs). The pBP2MB1-linker (a gift from Dr. Kensuke Kataoka, National Institute for Basic Biology, Japan) carries PAC under the control of the *MTT2* copper-inducible promoter (*Boldrin et al., 2008*) between SalI and XmaI sites. The MTT2-PAC cassette was excised from the vector and integrated into backbone vectors using T4 DNA ligase (New England BioLabs). The resulting vectors (pΔSPO11-PAC, pTOP2Gi-PAC, and

pASF1-GFP-PAC) were linearized with SacI plus KpnI before biolistic transformation into *Tetrahymena*.

## RT-PCR analysis

Total RNA was extracted from approximately $1 \times 10^6$ cells using RNAiso Blood (TaKaRa) and 0.5 µg was reverse transcribed using ProtoScript II Reverse Transcriptase (New England BioLabs). An *SPO11*-specific amplicon of ~700 bp was produced using the following primers: RT-*SPO11* forward – TGTTTAAATATTATTGCTTCAGC, reverse – ATAAACTCAGCATTTTCAATCC. The loading control was an *HSP70*-specific PCR product of ~500 bp, produced using the following primers: RT-*HSP70* forward – ATCTCTTGGGTAAGTTCAACC, reverse – TTGAAGACTTCTTCCAAAG.

## Pulsed-field gel electrophoresis

DNA plugs were made from $10^7$ cells (~10 ml cultured cells). A centrifuged cell pellet (100 µl) was resuspended in 250 µl 1% Low-Melt Agarose (Bio-Rad) at 42°C, and 80 µl was quickly loaded into each plug mold (Bio-Rad) on ice. After the agarose had solidified, plugs were transferred to a tube containing 400 µl LET buffer (0.5 M $Na_2$EDTA, 10 mM Tris-HCl, pH7.4) and kept on ice until samples had been collected for all time points. NDC solution (400 µl; LET buffer with 2% N-lauroylsarcosine [Sigma-Aldrich] containing 4 mg/ml proteinase K [Panreac AppliChem, Darmstadt, Germany]) was then added and the tubes were incubated overnight at 50°C. Plugs were then washed three times for 30 min with 1 M Tris-HCl (pH7.4) followed by three times for 30 min with TE buffer (1 mM $Na_2$EDTA, 10 mM Tris-HCl, pH7.4). PFGE was performed using a contour-clamped homogeneous electric field apparatus (CHEF-DR III System, Bio-Rad). Samples were separated at 2 V, 14°C in 0.85% Certified Megabase agarose (Bio-Rad) with $1\times$ TAE buffer (40 mM Tris, 40 mM acetic acid, 1 mM EDTA) for 74 hr: at a 96° angle for 24 hr with 1200 s pulses, a 100° angle for 24 hr with 1500 s pulses, and a 106° angle for 24 hr with 1800 s pulses. The CHEF DNA size markers (*Schizosaccharomyces pombe* and *S. cerevisiae* chromosomes, Bio-Rad) and Quick-Load 1 Kb Extend DNA Ladder (New England BioLabs) were used to size the DNA fragments. The gel was stained for 30 min in 10,000-fold diluted GelRed Nucleic Acid Gel Stain (Biotium, Fremont, CA) in distilled water and destained for 30 min in distilled water.

## Southern blotting and hybridization

DNA was transferred from pulsed-field gels onto Hybond N+ membrane (GE Healthcare, Little Chalfont, UK) with $20\times$ SSPE (3M NaCl, 20 mM EDTA, 154.8 mM $Na_2HPO_4$, 45.2 mM $H_6NaO_6P$, pH 7.4). The pMBR2 vector (NCBI: AF451863) carrying an 8.5 Kb fragment of the conserved internal region of Tlr elements (*Wuitschick et al., 2002*) was a gift from Dr Kathleen Karrer (Marquette University, USA). The Tlr sequence was excised from the vector with BamHI plus PstI (New England BioLabs) digestion, gel isolated, radioactively labeled by random priming using $^{32}$P-dATP (Hartmann Analytic, Braunschweig, Germany), and hybridized to germline DNA on the membrane. The signal was detected using Imaging Screen K (Bio-Rad) and scanned with a Typhoon 9200 image analyzer (GE Healthcare).

## Acknowledgements

We thank Kazufumi Mochizuki, Kensuke Kataoka, and Rachel Howard-Till for providing transformation vectors optimized for *Tetrahymena*; Kathleen Karrer for providing the pMBR2 vector; and Joel S Shore and Anura Shodhan for their technical help with PFGE. We would like to acknowledge Masaaki Iwamoto, Miao Tian, and Ali Emine Ibriam for their support. The research was funded by a Mahlke-Obermann Stiftung grant and the European Union's Seventh Framework Programme for research, technological development and demonstration (grant no. 609431) to TA; Grant-in-Aid funding from the Ministry of Education, Science, Sport and Culture (KAKENHI, 15K18475) to YF; grants from the Natural Sciences and Engineering Research Council of Canada (539509) and the Canadian Institutes for Health Research (MOP13347) to REP; and an Austrian Science Fund grant (P27313-B20) to JL.

## Additional information

### Funding

| Funder | Grant reference number | Author |
|---|---|---|
| Seventh Framework Programme | 609431 | Takahiko Akematsu |
| Japan Society for the Promotion of Science | 15K18475 | Yasuhiro Fukuda |
| Canadian Institutes of Health Research | MOP13347 | Ronald E Pearlman |
| Austrian Science Fund | P27313-B20 | Josef Loidl |
| Natural Sciences and Engineering Research Council of Canada | 539509 | Ronald E Pearlman |

The funders had no role in study design, data collection and interpretation, or the decision to submit the work for publication.

### Author contributions

TA, Conceptualization, Data curation, Formal analysis, Methodology, Writing—original draft; YF, Data curation, Formal analysis, Methodology; JG, Resources, Formal analysis; JSF, Resources, Formal analysis, Writing—review and editing; REP, Supervision, Writing—review and editing; JL, Resources, Supervision, Methodology, Writing—review and editing

### Author ORCIDs

Takahiko Akematsu, http://orcid.org/0000-0001-9396-0243

## Additional files

### Major datasets

The following previously published dataset was used:

| Author(s) | Year | Dataset title | Dataset URL | Database, license, and accessibility information |
|---|---|---|---|---|
| Griswold lab/Center for Reproductive Biology | 2006 | Microarray expression from isolated germ cell types | https://www.ncbi.nlm.nih.gov/geo/query/acc.cgi?acc=GSE2736 | Publicly available at the NCBI Gene Expression Omnibus (accession no: GSE2736) |

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
