## [Decision Letter]

Thank you for submitting your article "Topoisomerase II and Spo11 induce post-meiotic DNA double-strand breaks in *Tetrahymena thermophila* haploid pronuclei" for consideration by *eLife*. Your article has been reviewed by three peer reviewers, and the evaluation has been overseen by Reviewing Editor Kathleen Collins and Kevin Struhl as the Senior Editor. The following individuals involved in review of your submission have agreed to reveal their identity: Kazufumi Mochizuki (Reviewer #1); Elcin Unal (Reviewer #2).

The reviewers have discussed the reviews with one another and the Reviewing Editor has drafted this decision to help you prepare a revised submission.

Summary:

In this manuscript, Akematsu et al. provide compelling evidence that post-meiotic DNA double-strand breaks (DSBs) occur in *Tetrahymena*. They further demonstrate that the formation of the post-meiotic DSBs requires a germline-specific topoisomerase 2 (Top2G) as well as a meiotic endonuclease Spo11 and that the repair of these breaks facilitates chromatin remodeling and functional gamete formation. These are important observations because 1) post-meiotic DSB are suggested to occur in other eukaryotes including mammals and 2) no previous studies demonstrated the function of post-meiotic DSBs. A limited amount of additional work would improve the manuscript to be suitable for publication.

Essential revisions:

1) The imaging data can greatly benefit from quantitative analysis. It's unclear what percentage of the post-meiotic pronuclei display signs of DSBs and how this changes in *spo11* and *top2G* mutants. Similar analysis for Asf1-GFP and H3K56Ac will be helpful.

2) The mechanism of post-meiotic DSB repair is unknown, but correlates with H3K56ac. If possible, it would be informative to check markers specific for HR and/or NHEJ-directed repair such as Rad51, Ku, Mre11. This analysis will at least indicate whether these pathways are on during the time of post-meiotic DSB induction/repair.

3) The authors suggest that Top2G might be the B subunit for Spo11 in forming post-meiotic DSBs. I think it will be helpful if they determine whether Spo11 and Top2G interact with one another in a stage specific manner (i.e. in post-meiotic cells but not during meiotic prophase). Since they generated epitope-tagged alleles for both of these genes and since there is good time resolution between the occurrence of meiotic and post-meiotic breaks, a co-immunoprecipitation experiment in two different stages can be performed to address this question. Otherwise the speculation that Top2g could be an as-yet-undiscovered B-subunit for Spo11 seems ungrounded: Spo11 is a member of the TopVI family (heterotetramers) whereas one would assume Top2g (like other eukaryotic TOP2) is, a homodimer; it seems very hard to imagine that *Tetrahymena* Spo11 functions biochemically as a heterodimer with Top2g.

4) Have the authors checked whether the decline in γH2AX and the accompanying increase in H3K56ac occur in the unselected pronuclei in *∆ATG8* mutants? This could help distinguish whether there is a specific signal that directs DSB repair and chromatin remodeling to the selected pronuclei or whether the lack of nuclear elimination per se is sufficient for DSB repair to occur in the unselected pronuclei.

Figure 1/B – it would help to emphasize that the γH2Ax staining is *specific* to the pronuclei, and not observed in the MAC.

Observations of H2AX-GFP shown in Figure 2 says nothing about the specificity of the anti-gH2AX antibody. If I am wrong, the authors need to add some more explanation for why they think this experiment demonstrates the specificity. Otherwise, the authors may simply remove the figure.

5) Is it technically feasible to deplete Spo11 and/or Top2 activity only in the post-meiotic stage? This would directly test the requirement of Spo11/Top2 in PM-DSB formation independently of any role they may have in meiotic DSB formation (that indirectly influences PM-DSB formation).

6) It is stated that the spo11i strain has a defect in γH2Ax staining in the pronuclei (40-70%). However, the signal on the Southern in Figure 7 that is ascribed to PM-DSBs is actually greater in the spo11i mutant than the WT control. This doesn't seem to be consistent with the author's statements/hypotheses that such PM-DSBs are Spo11-dependent. Are they able to explain/clarify? The same concerns are present for Fig8E right-hand panel, which appears to show robust "PM-DSB" signal in the spo11i strain (equivalent to WT) – even though one assumes this DSB signal is arising from only a subpopulation of cells.

Is the PM-DSB signal observed in the Spo11i experiments dependent on Top2g? (i.e. is it possible to do a double mutant/knockdown at the same time?)

7) The model shown in Figure 11 gives me the impression that the deposition of H3 occurs only at close proximity to DSB sites, which the authors estimated to occur at only ~140 sites per 2xhaploid micronuclei genome. If this is true, it is rather very local remodeling event and it is difficult to imagine how DSBs promote gametogenesis, which probably requires more global chromatin remodeling. Because, in Figure 9, it seems H3K56ac are distributed homogeneously in the micronucleus, I believe it is more reasonable to consider that although the number of DSBs is limited, they induce global chromatin remodeling (may be irrespective of distances from DSBs). The authors may consider this point and tune their discussions.

---

## [Author Response]

*Essential revisions:*

*1) The imaging data can greatly benefit from quantitative analysis. It's unclear what percentage of the post-meiotic pronuclei display signs of DSBs and how this changes in spo11 and top2G mutants. Similar analysis for Asf1-GFP and H3K56Ac will be helpful.*

As recommended, new data have been included in the revised Figure 5, Figure 6, and 10C. These source data are available in the revised [Supplementary-material SD2-data], [Supplementary-material SD4-data], and [Supplementary-material SD5-data].

*2) The mechanism of post-meiotic DSB repair is unknown, but correlates with H3K56ac. If possible, it would be informative to check markers specific for HR and/or NHEJ-directed repair such as Rad51, Ku, Mre11. This analysis will at least indicate whether these pathways are on during the time of post-meiotic DSB induction/repair.*

We were able to visualize the *Tetrahymena* orthologs of human DNA-dependent protein kinase catalytic subunit (DNA-PKcs) and DNA recombinase Rad51 in the selected pronucleus but not in unselected pronuclei. These new findings have been included in the revisedFigure 3—figure supplement 1,Figure 3—figure supplement 2. As we showed that *Tetrahymena* pronuclei are in G_2_ (Supplementary file 1), it is conceivable that Rad51-dependent recombinational repair is mediated by the sister chromatid in the selected pronucleus. It is therefore possible that both NHEJ and recombinational repair contribute to PM-DSB repair.

*3) The authors suggest that Top2G might be the B subunit for Spo11 in forming post-meiotic DSBs. I think it will be helpful if they determine whether Spo11 and Top2G interact with one another in a stage specific manner (i.e. in post-meiotic cells but not during meiotic prophase). Since they generated epitope-tagged alleles for both of these genes and since there is good time resolution between the occurrence of meiotic and post-meiotic breaks, a co-immunoprecipitation experiment in two different stages can be performed to address this question. Otherwise the speculation that Top2g could be an as-yet-undiscovered B-subunit for Spo11 seems ungrounded: Spo11 is a member of the TopVI family (heterotetramers) whereas one would assume Top2g (like other eukaryotic TOP2) is, a homodimer; it seems very hard to imagine that Tetrahymena Spo11 functions biochemically as a heterodimer with Top2g.*

As recommended, we carried out a co-immunoprecipitation (co-IP) experiment in extracts from mating cells expressing Top2G-GFP and Spo11-FZZ, from two different stages: meiotic prophase (3 h) and post-meiotic stage (6 h;as shown in Figure 12). Spo11-FZZ was pulled down with anti-Flag antibody; however, Top2G-GFP did not co-IP with Spo11-FZZ. To account for the possibility that the in the native complex the C-terminal FZZ tag of Spo11-FZZ is hidden so that only free (i.e. unbound) Spo11-FZZ is pulled down, we carried out a reciprocal co-IP experiment using GFP-Trap. Western blotting detected Top2G-GFP in the IP fraction of post-meiotic cell lysate; however, Spo11-FZZ was not detected in the IP fractions of both the co-IP and control experiments (i.e. using lysates from cells not expressing Top2G-GFP). GFP-Trap consists of an anti-GFP single-domain antibody derived from alpaca and an affinity resin bead. Thus, Spo11-FZZ interacted non-specifically with components of GFP-Trap, and this could not be prevented by replacing the mild homogenization buffer (see Figure 12 figure legend) with the more stringent RIPA buffer (140 mM NaCl, 0.1% SDS, 1 mM EDTA, 1% Triton X-100, 0.1% sodium deoxycholate, 10 mM Tris-HCl, pH 8.0). It therefore remains unclear whether Top2g and Spo11 can interact. It is possible that epitope tagging of both Top2G and Spo11 prevents their interaction. It is also possible that the potential PM-DSB inducer requires other protein(s) to mediate the interaction between Top2G and Spo11 only at the post-meiotic stage.

For these technical reasons, the co-IP experiment was uninformative and has not been included in the manuscript. To circumvent these technical difficulties, an affinity purification mass spectrometry experiment is required to more fully elucidate the mechanism of PM-DSB induction. We are currently developing a protocol for proteomic analysis as, well as for ChIP-sequence analysis, of Top2G and Spo11 but these experiments are beyond the scope of this manuscript.

Author response image 1.Co-Ip analysis of Top2G-GFP and Spo11-FZZ.Mating Top2G-GFP and Spo11-FZZ expressing cells (10^7^ cells) at meiotic prophase (3 h) and the post-meiotic stage (6 h) were collected by centrifugation following pretreatment with 0.5 mM PMSF (Roche Diagnostics, Indianapolis, IN) for 30 min at 30°C. Cells were resuspended in 1 ml homogenization buffer (150 mM NaCl, 1% Triton X-100 [Sigma-Aldrich], 2 mM PMSF, and Complete Protease Inhibitor Cocktail [Roche Diagnostics]), homogenized by gentle pipetting on ice, and clarified by centrifugation at 10,000 *g* for 15 min. The resulting lysate was incubated with 25 μl GFP-Trap magnetic agarose beads (ChromoTek, Planegg-Martinsried, Germany) or 1 μg anti-FLAG antibody bound to 25 μl Protein G magnetic agarose beads (GE Healthcare) for 1 h at 4°C. After three washes with homogenization buffer, the beads were incubated with 30 μl SDS sample buffer at 98°C for 2 min to elute bound proteins. Samples (15 μl) of eluted proteins were separated by SDS-PAGE. WCL: whole cell lysate, IP, immunoprecipitation fraction.**DOI:**
http://dx.doi.org/10.7554/eLife.26176.023

*4) Have the authors checked whether the decline in γH2AX and the accompanying increase in H3K56ac occur in the unselected pronuclei in ∆ATG8 mutants? This could help distinguish whether there is a specific signal that directs DSB repair and chromatin remodeling to the selected pronuclei or whether the lack of nuclear elimination per se is sufficient for DSB repair to occur in the unselected pronuclei.*

We immunostained γH2AX and H3K56ac in the *ΔATG8-2* mutant, and these new data have been included in revised Figure 9. The selected pronucleus in the mutant crosses became H3K56 acetylated accompanied with H2AX dephosphorylation, as seen in the wild-type crosses. In addition, some persisting unselected pronuclei showed reduced γH2AX staining after the post-meiotic stage. Nevertheless, these pronuclei underwent neither H3K56 acetylation nor gametogenic mitosis, indicating that DNA repair and histone H3 transfer take place as a consequence of pronuclear selection.

*Figure 1/B – it would help to emphasize that the γH2Ax staining is specific to the pronuclei, and not observed in the MAC.*

We think that the reviewers meant to say Figure 2/B, since γH2AX staining is first shown in Figure 2. As suggested, we added the statement that “γH2AX foci […] do not appear in parental MACs throughout mating” in the revised Figure 2 legend to emphasize the specific localization of γH2AX in MIC-derived nuclei, including pronuclei.

*Observations of H2AX-GFP shown in Figure 2 says nothing about the specificity of the anti-gH2AX antibody. If I am wrong, the authors need to add some more explanation for why they think this experiment demonstrates the specificity. Otherwise, the authors may simply remove the figure.*

We agree with the reviewer’s comment; Figure 2 did not demonstrate the specificity of the anti-γH2AX antibody. Therefore, these images and the related text have been removed from the revised Figure 2 and main text, respectively. On the other hand, our observations with H2AX-GFP may be useful evidence that the sole *Tetrahymena* histone H2AX encoded by TTHERM_00790790 is indeed expressed in the pronuclei as well as in the meiotic prophase MIC. These data have therefore been moved to the revisedFigure 2—figure supplement 1 to support the conclusion of this section, namely that γH2AX immunostaining in the pronuclei is not due to cross-reaction of the antibody.

*5) Is it technically feasible to deplete Spo11 and/or Top2 activity only in the post-meiotic stage? This would directly test the requirement of Spo11/Top2 in PM-DSB formation independently of any role they may have in meiotic DSB formation (that indirectly influences PM-DSB formation).*

Although it is unclear whether all aspects of meiosis are normal in *TOP2gi* crosses, Southern blotting indicated that Top2G (at least) is not involved in meiotic DSB formation (Figure 7 and Figure 8). Moreover, 40–70% of the *SPO11i* crosses, in which Spo11 is re-expressed after meiotic prophase (Figure 6), formed PM-DSBs despite the absence of meiotic DSBs (Figure 7 and Figure 8). These results strongly suggest that PM-DSB formation is independent of meiotic DSBs. In addition, the mating process is unlikely to proceed to the post-meiosis stage if Top2G and Spo11 are required for meiotic DSB repair, as *RAD51* and *MRE11* mutants arrest at meiotic prophase (Lukaszewicz et al., 2010; Howard-Till et al., 2011). Of course, the best experiment to convince the reviewer would be to specifically deplete Top2G and Spo11 activities at the post-meiotic stage. However, it is technically difficult in *Tetrahymena* to restrict gene knockdown to a specific mating stage. So far, we have not succeeded in triggering RNAi specifically at post-meiosis using a variety of promoters (such as *TWI1, PDD1*, and *MTT2*).

*6) It is stated that the spo11i strain has a defect in γH2Ax staining in the pronuclei (40-70%). However, the signal on the Southern in Figure 7 that is ascribed to PM-DSBs is actually greater in the spo11i mutant than the WT control. This doesn't seem to be consistent with the author's statements/hypotheses that such PM-DSBs are Spo11-dependent. Are they able to explain/clarify? The same concerns are present for Fig8E right-hand panel, which appears to show robust "PM-DSB" signal in the spo11i strain (equivalent to WT) – even though one assumes this DSB signal is arising from only a subpopulation of cells.*

*Is the PM-DSB signal observed in the Spo11i experiments dependent on Top2g? (i.e. is it possible to do a double mutant/knockdown at the same time?)*

It is true that in the *spo11i* cells, in which RNAi is attenuated at late stages by 40–70% (as judged by the percentage of pronuclei showing γH2AX staining), the PM-DSB signal would be expected to be only about half the strength of the wild-type PM-DSB signal. However, RNAi attenuation varied greatly between experiments, and signal quantification and comparison between gels is difficult. (Note that the rDNA loading control on the *SPO11i* gel indicates that more DNA was loaded or transferred, which may [partially] explain the somewhat stronger than expected PM-DSB signal in *SPO11i* cells; Figure 7). In the *SPO11i::ΔATG8-2* mutant (see Figure 8), the reduction compared with *SPO11* + *ΔATG8-2* is more evident. As suggested by the reviewers, generation of a triple *TOP2gi::SPO11i::ΔATG8-2*mutant showed that the remaining PM-DSBs in *SPO11i* are dependent on *TOP2g*.

*7) The model shown in Figure 11 gives me the impression that the deposition of H3 occurs only at close proximity to DSB sites, which the authors estimated to occur at only ~140 sites per 2xhaploid micronuclei genome. If this is true, it is rather very local remodeling event and it is difficult to imagine how DSBs promote gametogenesis, which probably requires more global chromatin remodeling. Because, in Figure 9, it seems H3K56ac are distributed homogeneously in the micronucleus, I believe it is more reasonable to consider that although the number of DSBs is limited, they induce global chromatin remodeling (may be irrespective of distances from DSBs). The authors may consider this point and tune their discussions.*

The reviewers are correct. The sparse PM-DSBs would not be sufficient for global histone replacement if this replacement occurred only in the vicinity (up to 7 kb as is the case in humans) (Goldstein et al., 2013; Li and Tyler, 2016) of DSBs. Therefore, either PM-DSB repair triggers histone replacement at distant sites by an as yet unknown mechanism, or nuclear dedifferentiation in *Tetrahymena* requires only local histone replacement events. We have added a discussion of these possibilities to the manuscript.